



# Room temperature hyperpolarization of polycrystalline samples with optically polarized triplet electrons: Pentacene or Nitrogen-Vacancy center in diamond?

Koichiro Miyanishi[1], Takuya F. Segawa[2,3], Kazuyuki Takeda[4], Izuru Ohki[5], Shinobu Onoda[6, 7], Takeshi Ohshima[6, 7], Hiroshi Abe[6, 7], Hideaki Takashima[8], Shigeki Takeuchi[8], Alexander I. Shames[9], Kohki Morita[5], Yu Wang[4], Frederick T.-K. So[2,6], Daiki Terada[2,6], Ryuji Igarashi[6,7,10], Akinori Kagawa[1,10,11], Masahiro Kitagawa[1,11], Norikazu Mizuochi[5], Masahiro Shirakawa[2,6], and Makoto Negoro[6,10,11]

[1]Graduate School of Engineering Science, Osaka University, Toyonaka, Osaka 560-8531, Japan
[2]Department of Molecular Engineering, Graduate School of Engineering, Kyoto University, Nishikyo-Ku, Kyoto 615-8510, Japan
[3]Laboratory for Solid State Physics, ETH Zurich, 8093 Zurich, Switzerland
[4]Division of Chemistry, Graduate School of Science, Kyoto University, Kyoto 606-8502, Japan
[5]Institute for Chemical Research, Kyoto University, Gokasho, Uji, Kyoto, 611-0011, Japan
[6]Institute for Quantum Life Science, National Institutes for Quantum and Radiological Science and Technology, 4-9-1, Anagawa, Inage-Ku, Chiba 263-8555, Japan
[7]Takasaki Advanced Radiation Research Institute, National Institutes for Quantum and Radiological Science and Technology, 1233 Watanuki, Takasaki, Gunma 370-1292, Japan
[8]Department of Electronic Science and Engineering, Kyoto University, Nishikyo-ku, Kyoto 615-8510, Japan
[9]Department of Physics, Ben-Gurion University of the Negev, 8410501 Beer-Sheva, Israel
[10]JST, PRESTO, Kawaguchi, Japan
[11]Quantum Information and Quantum Biology Center, Institute for Open and Transdisciplinary Research Initiatives, Osaka University, Japan

**Correspondence:** Koichiro Miyanishi (miyanishi@qi.mp.es.osaka-u.ac.jp), Takuya F. Segawa (segawat@ethz.ch), Makoto Negoro (negoro@qiqb.otri.osaka-u.ac.jp)

**Abstract.** We demonstrate room-temperature $^{13}$C hyperpolarization by dynamic nuclear polarization (DNP) using optically polarized triplet electron spins in two polycrystalline systems: pentacene-doped [carboxyl-$^{13}$C] benzoic acid and microdiamonds containing NV$^-$ centers. For both samples, the integrated solid effect (ISE) is used to polarize the $^{13}$C spin system in magnetic fields of 350-400 mT. In the benzoic acid sample, the $^{13}$C spin polarization is enhanced up to 0.12 % through direct electron-to-$^{13}$C polarization transfer without performing dynamic $^1$H polarization followed by $^1$H-$^{13}$C cross polarization. In addition, ISE has been successfully applied for the first time to polarize naturally abundant $^{13}$C spins in a microdiamond sample to 0.01 %. To characterize the buildup of the $^{13}$C polarization, we discuss the efficiencies of direct polarization transfer between the electron and $^{13}$C spins as well as that of $^{13}$C–$^{13}$C spin diffusion, examining various parameters which are beneficial or detrimental for successful bulk dynamic $^{13}$C polarization.





## 1 Introduction

Nuclear magnetic resonance (NMR) spectroscopy and magnetic resonance imaging (MRI) are powerful tools covering fields from physics, chemistry, biology to medicine. However, the poor sensitivity remains the Achilles heel to all magnetic resonance experiments. Dynamic nuclear polarization (DNP), a technique to transfer spin polarization from electrons to nuclei, has extensively been studied since its early discovery (Overhauser, 1953) and has seen an impressive revival in the current
century (Ardenkjær-Larsen et al., 2003; Maly et al., 2008; Lesage et al., 2010; Nelson et al., 2013). In DNP using unpaired electrons as the sources of polarization, the polarization enhancement factor $\epsilon$ is limited to $\gamma_e/\gamma_n$, where $\gamma_{e(n)}$ are the gyromagnetic ratios of the electron (nuclear) spins. To attain nuclear polarization of as high as $\sim 10\,\%$, DNP needs to be performed at very low temperature (<20 K) and in high magnetic fields (>3 T).

Conversely, the spins of optically polarized electrons can have much higher polarization than the thermal equilibrium value.
DNP using such attractive sources of polarization leads to nuclear hyperpolarization beyond the limit of DNP using thermal electron polarization. Moreover, the spin polarization of the optically polarized electrons does not depend on the temperature and the magnetic field. It follows that nuclear hyperpolarization is expected even at ambient temperatures and in relatively low magnetic fields. In times of global liquid helium shortage, this could become an elegant method without requiring liquid helium for sample cooling. DNP using optically-polarized electron spins in the triplet state (Henstra et al., 1990; Stehlik and
Vieth, 1992), triplet-DNP, was originally demonstrated in a single crystal of naphthalene doped with pentacene, and this DNP technique using pentacene achieved [1]H polarization of 34 % at room temperature in 0.4 T (Tateishi et al., 2014). DNP using pentacene has been extended to polycrystalline samples (Takeda et al., 2001), and dissolution DNP using pentacene-doped powder samples (Negoro et al., 2018; Kagawa et al., 2019) has been implemented. Furthermore, the range of molecules, hyperpolarized by triplet-DNP, was successfully expanded (Kagawa et al., 2018; Tateishi et al., 2019; Nishimura et al., 2020).
In these DNP experiments, the integrated solid effect (ISE) was used as a method for transferring the polarization. ISE employs microwave irradiation and external magnetic-field sweep, so that the Hartmann-Hahn matching is realized between the electron spins in the rotating frame and the nuclear spins in the laboratory frame (Henstra et al., 1990).

In proton-rich organic solids or frozen solutions, a demonstrated strategy to polarize dilute or low-$\gamma$ nuclear spins is first to transfer the electron polarization to the protons, and then let the protons undergo spin diffusion to spatially transport the
enhanced polarization away from the source electron spins, and finally to perform local polarization transfer from the protons to the target spin species. The protons in rigid solids are known to be efficient carriers of spin polarization. However, when, as is often the case, it is not the protons but other dilute or low-$\gamma$ spin species that are of NMR-spectroscopic interest, [1]H hyperpolarization can be costly, because the relatively large heat capacity of the [1]H reservoir consumes considerable resources of electron polarization. Then, a question arises: would direct dynamic polarization of dilute/low-$\gamma$ spins using the electrons
in the triplet state be feasible without the protons being involved in the polarization-buildup process, and if it is, how efficient would that be? In this context, dynamic polarization of dilute/low-$\gamma$ spins in proton-free inorganic solids is also of interest.

Here, we study dynamic [13]C polarization using optically polarized electron spins in the triplet state through direct electron-to-[13]C polarization transfer, i.e., without being mediated by the [1]H spins. In this work, we deal with pentacene doped into a





host matrix and nitrogen-vacancy ($NV^-$) color centers in diamond. In both cases, the optically-polarized triplet electron spins

are the source for DNP. In the case of the $NV^-$ center, laser irradiation excites the electronic state from the ground triplet state $^3A_2$ to the excited triplet state $^3E$ (Fig. 1), which then undergoes intersystem crossing (ISC) to the singlet state $^1A_1$. Importantly, this transition is highly spin selective; while the $m_S = 0$ state has a low ISC probability, $m_S = \pm1$ state has a high ISC probability. The singlet state $^1A_1$ further decays into a meta-stable singlet state $^1E_1$, which preferentially decays into the $m_S = 0$ spin sub-level of the ground triplet state $^3A_2$, hyperpolarizing the $m_S = 0$ state. In the case of pentacene, the ground

singlet state $S_0$ is photo-excited to the excited singlet state $S_1$. The subsequent transition to the triplet state $T_3$ is caused by ISC, and then to the lowest triplet state $T_1$ by internal conversion, where the spin-orbit coupling selectively populates the triplet sublevels. The triplet population depends on the host molecule and is independent of temperature and magnetic field.

Even though the relevant electron spins are in the triplet state ($S = 1$) for both the pentacene molecules and the $NV^-$ centers, they differ in many aspects. For the $NV^-$ centers, the ground state is the triplet state, while in the case of pentacene, the ground

state is the singlet state ($S = 0$). The zero-field splitting (ZFS) parameters $D$ and $E$ are 2870 MHz and $\sim0$ MHz for the $NV^-$ center and $\sim$1350 MHz and $\sim$42 MHz for pentacene in the excited state, the latter of which may vary depending on the host molecule (Yu et al., 1984). The electronic structure for these systems are shown in Fig. 1 (Takeda, 2009; Doherty et al., 2013; Rogers et al., 2008; Acosta et al., 2010) and the differences between these two systems are summarized in Table 1.

DNP for a bulk ensemble of $^{13}$C nuclear spins using the $NV^-$ centers in a diamond single crystal was first demonstrated

at cryogenic temperature in a magnetic field of 9.4 T (King et al., 2010). Microwave-free optical hyperpolarization at room temperature, where polarization transfer by excited-state level anticrossing at 50 mT was followed by sample shuttling to a magnetic field of 4.7 T for NMR detection, was also demonstrated (Fischer et al., 2013). The method was generalized to a broader range of magnetic fields and correspondingly different orientations of the $NV^-$ center (Álvarez et al., 2015). For a single crystal of diamond, the $^{13}$C polarization of 6 % at room temperature has been achieved via the combination of the

thermal mixing and the solid effects (King et al., 2015). Recently, DNP using $NV^-$ in powdered microdiamonds in a magnetic field of as low as ca. 30 mT has been reported by Ajoy et al., who took advantage of the reduced width of the anisotropic electron paramagnetic resonance (EPR) powder pattern of the $NV^-$ centers (Ajoy et al., 2018a, b).

In the following, we report on optical DNP of $^{13}$C spins in microdiamonds containing $NV^-$ centers and in pentacene-doped [carboxyl-$^{13}$C] benzoic acid (PBA) microcrystals. While for the diamond samples the $^{13}$C isotope is naturally abundant,

the carboxyl carbons are isotopically $^{13}$C-enriched in PBA microcrystals. We study the behavior of the buildup of the $^{13}$C polarization in terms of the efficiency of the polarization transfer from the electron to nuclear spins and that of $^{13}$C spin diffusion. As demonstrated below, we obtained $^{13}$C polarization of 0.01 % in the microdiamonds, and 0.12 % in PBA at room temperature in a magnetic field of $\sim$0.4 T by the ISE scheme.

In a slightly different context, both the pentacene and $NV^-$ systems can be used for single-spin optically-detected magnetic

resonance (ODMR). Pentacene in $p$-terphenyl was the first single molecule, which was detected by magnetic resonance (Köhler et al., 1993; Wrachtrup et al., 1993). While these experiments were performed at cryogenic temperatures, the $NV^-$ center in diamond opened the door for single-spin experiments at room temperature (Gruber et al., 1997; Jelezko et al., 2004). In this article, we will focus on the conventional induction detection of EPR/NMR on ensembles of spins.





**Table 1.** Differences between the two samples, NV$^-$ in microdiamonds and pentacene in [carboxyl-$^{13}$C] benzoic acid ($\alpha^{13}$CBA)

| Sample | NV$^-$ in microdiamonds | Pentacene in $\alpha^{13}$CBA |
|---|---|---|
| ZFS $D$ [MHz] | 2870 | 1350 |
| ZFS $E$ [MHz] | $\sim 0$ | -42 |
| electron spin | defect in crystal | 'doped molecule' |
| ground state | triplet state | singlet state |
| meta-stable state | singlet state | triplet state |
| average $^{13}$C distance $r_{\mathrm{C}}$ ($\rho_{\mathrm{C}}^{1/3}$) [Å] | $\sim 8.0$ | $\sim 5.4$ |

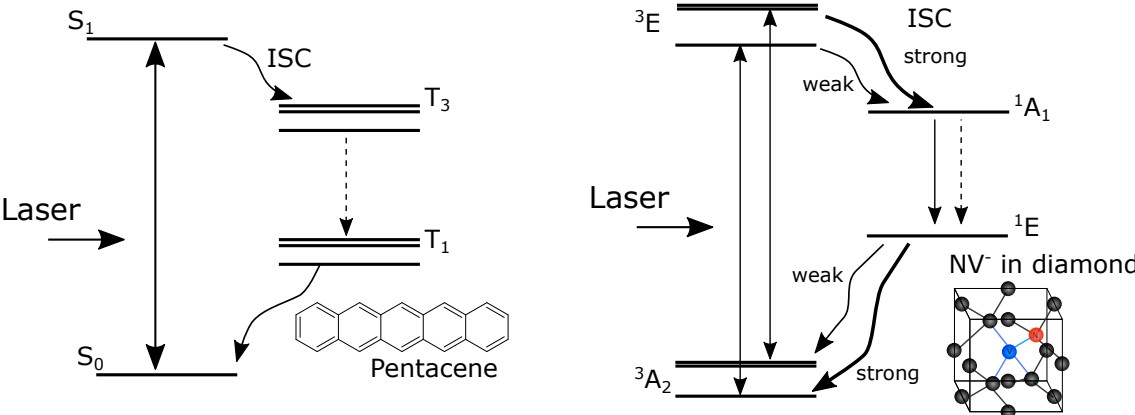

**Figure 1.** Schematic diagrams of the electronic structure for NV$^-$ and pentacene. The optical transitions are denoted by solid straight arrows, the transitions caused by intersystem crossing are denoted by wavy arrows and the transitions by internal conversion are denoted by dashed straight arrows.

## 2 Theory

### 2.1 Dynamic Nuclear Polarization using the Integrated Solid Effect

Let us suppose that pulsed laser irradiation creates hyperpolarized electron spins in the triplet state. In ISE, microwave irradiation and magnetic-field sweep are applied in such a way that the resonances of the individual electron spin packets between two triplet sublevels are adiabatically swept over and the Hartmann-Hahn condition is fulfilled at some point during the sweep, causing the contact between the electron spins in the rotating frame and the nuclear spins in the laboratory frame (Henstra et al., 1990). The Hartmann-Hahn condition is expressed as

$$\omega_{\mathrm{eff},e} = \omega_{0,\mathrm{C}}. \tag{1}$$

Here, $\omega_{\mathrm{eff},e}$ represents the electron nutation frequency around the effective field, and is given by $\sqrt{\omega_{1,e}^2 + (\Delta\omega_e)^2}$, where $\omega_{1,e}$ is the microwave intensity and $\Delta\omega_e$ is the time-dependent resonance-offset frequency. $\omega_{0,\mathrm{C}}$ is the Larmor precession frequency



of the nuclear spin species of interest, which is $^{13}$C in the present case. The exchange of the spin states and thereby of spin po-
larization is driven by the dipolar interaction between them through the same mechanism as that of cross polarization (Mehring,
1983; Peng et al., 2006). Thus, the source electron spins need to be locked along the effective field for polarization transfer to
take place. This is ensured when the adiabatic sweep starts from far-off resonance, so that the effective field is initially aligned
nearly along the static field and gradually tilts.

Since the polarization transfer is driven by the local dipolar interaction, direct hyperpolarization is limited to those nuclei
which happen to be located in the vicinity of the electrons. Nevertheless, such locally hyperpolarized $^{13}$C spin state can be
transported away by spin diffusion. Thus, by repeating the ISE sequence as described in Fig. 2, the $^{13}$C polarization can be
accumulated until the DNP buildup and the nuclear spin-lattice relaxation balance and a steady state is established.

In general, both the direct electron-to-$^{13}$C polarization transfer by ISE and spin diffusion among the $^{13}$C spins contribute to
the overall buildup of the bulk $^{13}$C polarization. Conversely, when the $^{13}$C polarization is smoothed out within the interval of the
repeated ISE sequences, the buildup behavior of the bulk $^{13}$C spin polarization $P_{\mathrm{C}}$ becomes independent of the spin-diffusion
rate. In this *rapid-diffusion limit* (Takeda, 2009), the time evolution of $P_{\mathrm{C}}$ is governed by

$$\frac{d}{dt}P_{\mathrm{C}} \simeq R\xi\frac{\eta_{\mathrm{t}}\rho_e}{\rho_{\mathrm{C}}}(\eta_{\mathrm{p}}\overline{P_e} - P_{\mathrm{C}}) - \frac{1}{T_{1,\mathrm{C}}}P_{\mathrm{C}}, \tag{2}$$

where $R$ is the repetition rate of the ISE sequence, $\eta_{\mathrm{t}}$ is the fraction of the triplet electrons in two of the triplet sublevels and is
obtained as $\frac{2}{3-P_e}$ by considering that the populations in the $m_S = +1$ and the $m_S = -1$ states are the same, $\xi$ is the exchange
probability, i.e., the probability of the spin states being exchanged between the electron and the nuclear spins during a single
ISE sequence, $\rho_e$ is the density of the electrons in the triplet state, $\rho_{\mathrm{C}}$ is the density of $^{13}$C spins, and $T_{1,\mathrm{C}}$ is the longitudinal
relaxation time of the $^{13}$C spins. $\eta_{\mathrm{p}}(\le 1)$, called the active spin-packet fraction, is introduced to deal with such situations
that not all but a part of the electron spin packets participate in the process of polarization transfer. This is indeed the case
for polycrystalline samples, where the anisotropy of the ZFS tensor causes significant broadening to such an extent that the
resonance line can only partly be excited. $\overline{P_e}$ is the electron polarization between the two of triplet sublevels averaged over the
ISE sweep time $t_{\mathrm{MW}}$, which is given as

$$\overline{P_e} = \frac{1}{t_{\mathrm{MW}}}\int\limits_0^{t_{\mathrm{MW}}} P_e \exp(-t/T_{1,e})dt. \tag{3}$$

Here, $P_e$ is the initial electron polarization between the relevant two triplet sublevels. $T_{1,e}$ is the time constant introduced to
take either spin-lattice relaxation or lifetime decay of the triplet state into account.

Neglecting the thermal polarization of $^{13}$C spins, we obtain the solution of this differential equation as

$$P_{\mathrm{C}}(t) = P_{\mathrm{fin}}(1 - \exp(-t/T_b)), \tag{4}$$

$$\frac{1}{T_b} = \frac{R\xi\eta_{\mathrm{t}}\rho_e}{\rho_{\mathrm{C}}} + \frac{1}{T_{1,\mathrm{C}}}, \quad P_{\mathrm{fin}} = \frac{T_{1,\mathrm{C}}\eta_{\mathrm{p}}\overline{P_e}}{\frac{\rho_{\mathrm{C}}}{R\xi\eta_{\mathrm{t}}\rho_e} + T_{1,\mathrm{C}}} \tag{5}$$





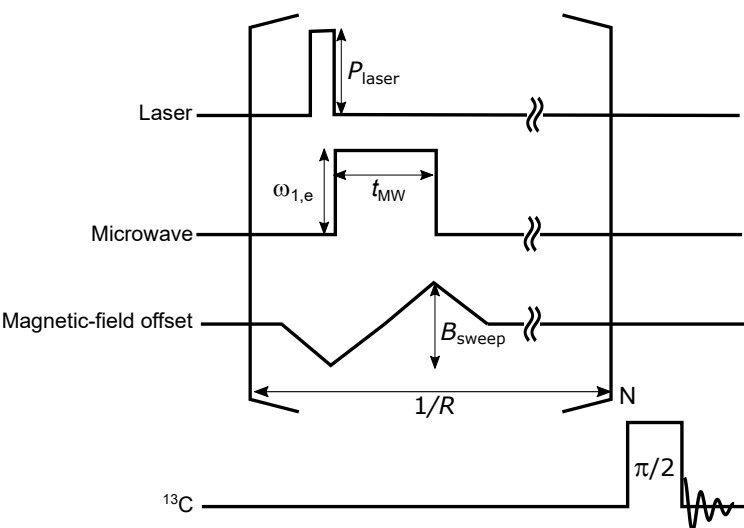

**Figure 2.** The pulse sequence for ISE followed by NMR detection. A laser pulse is used for photoexcitation and optical polarization of the triplet electron spins. Then, a microwave pulse is applied together with magnetic-field sweep. The ISE pulse sequence is repeated before the enhanced $^{13}$C magnetization is detected by applying a radiofrequency $\pi/2$ pulse. During the acquisition of the $^{13}$C signal, $^1$H decoupling is to be applied in the case of the benzoic acid sample. The definitions of the symbols used in the figure are as follows. $P_{\mathrm{laser}}$: laser-beam intensity, $\omega_{1,e}$: microwave intensity, $t_{\mathrm{MW}}$: microwave-pulse width, $B_{\mathrm{sweep}}$: field-sweep width, $R$: ISE repetition rate, $N$: ISE repetition number.

Here, $P_{\mathrm{fin}}$ is the maximum attainable nuclear spin polarization and $T_b$ is the DNP buildup time constant. We note that the initial buildup rate, given as

$$\frac{d}{dt}P_{\mathrm{C}}(t=0) = R\frac{\rho_e}{\rho_{\mathrm{C}}}\eta_{\mathrm{t}}\eta_{\mathrm{p}}\xi\overline{P_e}, \quad (6)$$

is independent of the longitudinal relaxation time $T_{1,\mathrm{C}}$.

In the following, we will use Eqs. (4)-(6) for data analysis.

The rapid-diffusion limit can be made valid by performing experiments with a sufficiently slow repetition rate $R$. Buildup experiments with the slow repetition rate, albeit being not advantageous for attaining the highest possible nuclear polarization, give helpful data that allows us to characterize the parameters.

## 3 Experimental

### 3.1 Sample preparation

We used the following three samples: (i) 70 mg of 500 $\mu$m microdiamonds, with $8.9 \times 10^{17}$ cm$^{-3}$ (5 ppm in atomic ratio) NV$^-$ concentration and $4.6 \pm 0.1 \times 10^{18}$ cm$^{-3}$ (26 ppm) P1 center concentration. So-called 'P1 centers' are another type of electron spin defects in diamond with $S = 1/2$, which are located on single substitutional nitrogen atoms. (ii) 5.6 mg of 100 nm nanodiamonds, with $5.3 \times 10^{17}$ cm$^{-3}$ (3 ppm) NV$^-$ concentration, $1.2 \times 10^{19}$ cm$^{-3}$ (70 ppm) P1 center concentration and a



concentration of $3.1 \times 10^{19}$ cm$^{-3}$ (177 ppm) for the total $S = 1/2$ electron spins (P1 centers, dangling bonds, etc.). 100 nm sized nanodiamonds "MICRON+ MDA M0.10" were purchased from Element Six, UK and electron irradiated at a fluence of $10^{19}$ e$^{-}$/cm$^2$ at room temperature to form vacancies. To create the NV$^{-}$ centers, the nanodiamond sample were annealed at

800°C under vacuum ($< 10^{-4}$ Pa). The temperature was increased to 400°C over 1 h and kept for 4 h ('baking'). After that, the temperature was increased to 800°C over 11 h, where the sample was annealed for 2 h. The temperature was decreased to 350°C in 1 h and then down to room temperature. These "dark" nanodiamonds (due to amorphous sp$^2$ carbon on the surface) were "made white" by oxidation in air at 550 °C for 2 h, followed by boiling acid cleaning in H$_2$SO$_4$/HNO$_3$ (3:1, 125 °C) for 3 d (Terada et al., 2019). Both diamond samples contain 1.1 % natural abundance $^{13}$C. (iii) 4 mg of PBA powder, with

0.04 mol% ($2.6 \times 10^{18}$cm$^{-3}$) concentration of pentacene. The [carboxyl-$^{13}$C] benzoic acid were purchased from Cambridge Isotope Laboratories. The characteristics of the samples are summarized in Table 3. All samples were packed in a glass tube and the figures of these samples are shown in supplementary material. Pictures of all three samples can be found in the SI.

**Table 2.** Summary of sample conditions and the optimised parameters of the DNP experiments.

| Sample | nanodiamonds | microdiamonds | Pentacene-doped benzoic acid |
|---|---|---|---|
| density of NV$^{-}$ | $5.3 \times 10^{17}$ cm$^{-3}$ (3 ppm) | $8.9 \times 10^{17}$ cm$^{-3}$ (5 ppm) | - |
| density of P1 | $1.2 \times 10^{19}$ cm$^{-3}$ (70 ppm) | $4.6 \pm 0.1 \times 10^{17}$ (26 ppm) | - |
| density of pentacene | - | - | $2.6 \times 10^{18}$cm$^{-3}$ (0.04 mol%) |
| particle size | $\sim$100 nm | $\sim$500 $\mu$m | <100 $\mu$m |
| $B_{\mathrm{sweep}}$ [mT] | (no DNP performed) | 6 | 30 |
| $t_{\mathrm{MW}}$ [$\mu$s] | (no DNP performed) | 1250 | 30 |

## 3.2 Experimental setup

The procedure and experimental setup of triplet-DNP used in this work are similar to those in Ref. (Tateishi et al., 2014).

A solid-state laser with a wavelength of 527 nm, pulse length of 200 ns and pulse energy of 30 mJ was used as a light source for the excitation of NV$^{-}$ centers and a dye laser with a wavelength of 594 nm, pulse length of 200 ns and pulse energy of 6 mJ was used as a light source for excitation of pentacene. A static magnetic field from 0.3 T to 0.5 T was generated by an electromagnet. All experiments were carried out at room temperature.

The EPR experiments were done with a home-built spectrometer, similar to Ref. (Yap et al., 2015), where an RF pulse

at 400 MHz generated by arbitrary waveform generator was converted to 11.6 GHz in a superheterodyne architecture. The microwave pulse was amplified to 1 W and led into the cavity.



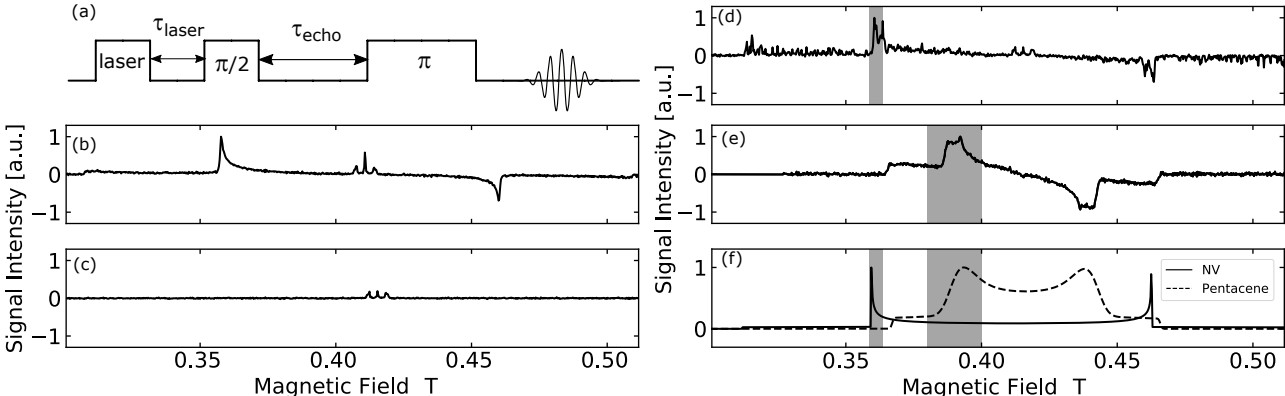

**Figure 3.** EPR powder spectrum of all samples obtained by simulation and experiment. (a) EPR echo pulse sequence used in the measurement. $\tau_{\text{laser}}$ is the delay time used for the lifetime decay measurement and $\tau_{\text{echo}}$ is the echo time. (b) Optically-polarized EPR powder spectrum of nanodiamonds with MW frequency of 11.52 GHz. (c) EPR powder spectrum of microdiamonds with MW frequency of 11.63 GHz without the laser pulse irradiation. (d) Optically-polarized EPR powder spectrum of microdiamonds with MW frequency of 11.63 GHz. (e) EPR powder spectrum of powder BA doped with 0.04 mol% pentacene with MW frequency of 11.66 GHz. (f) Simulated EPR powder spectra of thermal state triplet electron in NV center and pentacene at 11.6 GHz. The shaded area is the magnetic field sweep range $B_{\text{sweep}}$ for each system.

## 4 Results and discussion

### 4.1 EPR measurements

Using the pulse sequence depicted in Fig. 3(a), magnetic-field dependences of the amplitude of the spin echoes of the electrons
in the triplet state were measured for the three samples described above.

Fig. 3(b) shows the optically-polarized NV$^-$ powder spectrum of the nanodiamonds. The echo time $\tau_{\text{echo}}$ was set to 400 ns and the measurement took 7 h. The line exhibited a dipolar powder pattern (Pake pattern), where the two 'horns' are separated by the zero-field splitting parameter $D$ and the two 'shoulders' by $2D$. The horns come from the crystallites with such orientation that the N-V axis is perpendicular to the magnetic field, whereas the shoulders correspond to those in which the
N-V axis is along the field. The result that one of the transitions is inverted is ascribed to the optical polarization mechanism, which exclusively populates the electron spins of the ground triplet state in the NV center into the $m_S = 0$ state. To the best of our knowledge, this is the first report on the full powder EPR spectrum of optically-polarized nanodiamonds observed with the conventional (i.e., not optical) detection. Using the EPR spectrum in Fig. 3(b), we estimated the initial electron polarization $P_e^{\text{NV}}$ of the NV$^-$ center between the $m_S = 0$ and the $m_S = -1$ states right after laser irradiation to be 13 % from the
ratio of the area intensity of the EPR line of the NV$^-$ centers integrated over the field-sweep range to the area intensity of the EPR lines of all $S = 1/2$ electron spins (177 ppm). This corresponds to an enhancement of roughly 140 fold compared to Boltzmann polarization of 0.093 %. Importantly, an optically-polarized NV$^-$ signal was not seen for the "dark" nanodiamond





powder, before the chemical surface cleaning. We assume that the visible laser light was absorbed on the particle surface. The three lines in the center of the field (around $g = 2$) do not stem from the $NV^-$ center but from the P1 centers and other $S = 1/2$ electron spin defects. The hyperfine structure observed arises from the coupling of the electron spin to the adjacent $^{14}N$ nucleus in the P1 centers. This signal is not optically-polarized and also present 'in the dark' as shown in Fig. 3(c) for the case of microdiamonds. The optically-excited NV spectrum from microdiamonds in Fig. 3(d) does not show a smooth powder pattern but rather partially averaged crystalline pattern due to the average diameter of the particles was around 500 $\mu$m and the number of the microdiamonds in the sample tube was only $\simeq 200$. The echo time $\tau_{\mathrm{echo}}$ was set to 2.3 $\mu$s and the measurement took 3 h.

Fig. 3(e) shows the powder spectrum of the photo-excited triplet state of pentacene. In the measurement, the echo time $\tau_{\mathrm{echo}}$ was set to 2.3 $\mu$s and the measurement took 5 h. The shape is also a dipolar powder pattern, where the two 'horns' are separated by the zero-field splitting parameter $D$ and the two 'shoulders' by $2D$. The zero-field splitting parameter $D$ for pentacene is only about half of that for $NV^-$, which explains half the powder "linewidth" of pentacene compared to $NV^-$. In the case of pentacene, the part of the 'horn' is broadened due to the finite ZFS parameter $E$, which is zero for $NV^-$. Fig. 3(f) shows EPR spectra of the thermally populated $NV^-$ center and pentacene simulated using EasySpin, a Matlab package (Stoll and Schweiger, 2006).

## 4.2 Hyperpolarized $^{13}C$ NMR

### 4.2.1 $NV^-$-containing microdiamonds

We performed hyperpolarization of the $^{13}C$ spins in microdiamonds using the ISE pulse sequence in a magnetic field of 0.36 T, which corresponds to the position of the low-field 'horn' in the spectrum of Fig. 3(d). To find the experimental parameters that maximize the efficiency of DNP, we varied the range of the magnetic field sweep $B_{\mathrm{sweep}}^{\mathrm{NV}}$, the width $t_{\mathrm{MW}}^{\mathrm{NV}}$ and the amplitude $\omega_{1,e}^{\mathrm{NV}}$ of the microwave pulse, and examined the enhanced $^{13}C$ magnetizations. As demonstrated in the upper part of Fig. 4(a)-(c), the optimal conditions were found to be $B_{\mathrm{sweep}}^{\mathrm{NV}} \simeq 5$ mT, $t_{\mathrm{MW}}^{\mathrm{NV}} \simeq 1250$ $\mu$s, and $\omega_{1,e}^{\mathrm{NV}} \simeq 3.74$ MHz.

Then, adopting these parameters, we performed dynamic $^{13}C$ polarization by repeating the ISE sequence at a rate of 100 Hz for 240 s, measured $^{13}C$ NMR at a Larmor frequency $\omega_{0,\mathrm{C}}$ of 3.85 MHz, and successfully obtained a hyperpolarized $^{13}C$ spectrum of the microdiamonds, as demonstrated in Fig. 5(a). The result of the identical $^{13}C$ measurement except that DNP was not performed, also plotted in Fig. 5(a) for comparison, did not show any appreciable sign of the signal above the noise level. Fig. 5(b) shows buildup curves of $^{13}C$ polarization with various ISE repetition rates $R$ ranging from 10 Hz to 100 Hz. For $R$ of up to 60 Hz, the buildup rate and the finally attained $^{13}C$ polarization increased with $R$, whereas they saturated for $R > 60$ Hz. The maximum $^{13}C$ polarization was 0.01 %, corresponding to 324-fold enhancement of $^{13}C$ polarization compared to that in thermal equilibrium.

Fig. 5(c) shows the profile of $^{13}C$ depolarization after hyperpolarization, from which the time constant $T_{1,\mathrm{C}}^{\mathrm{Dia}}$ of $^{13}C$ longitudinal relaxation was determined to be $99 \pm 14$ s. Here, $^{13}C$ relaxation is mainly caused by the P1 centers (Ajoy et al., 2019) creating fluctuating local fields at the $^{13}C$ atomic sites. Fig. 5(d) compares the magnetic-field dependences of the enhanced





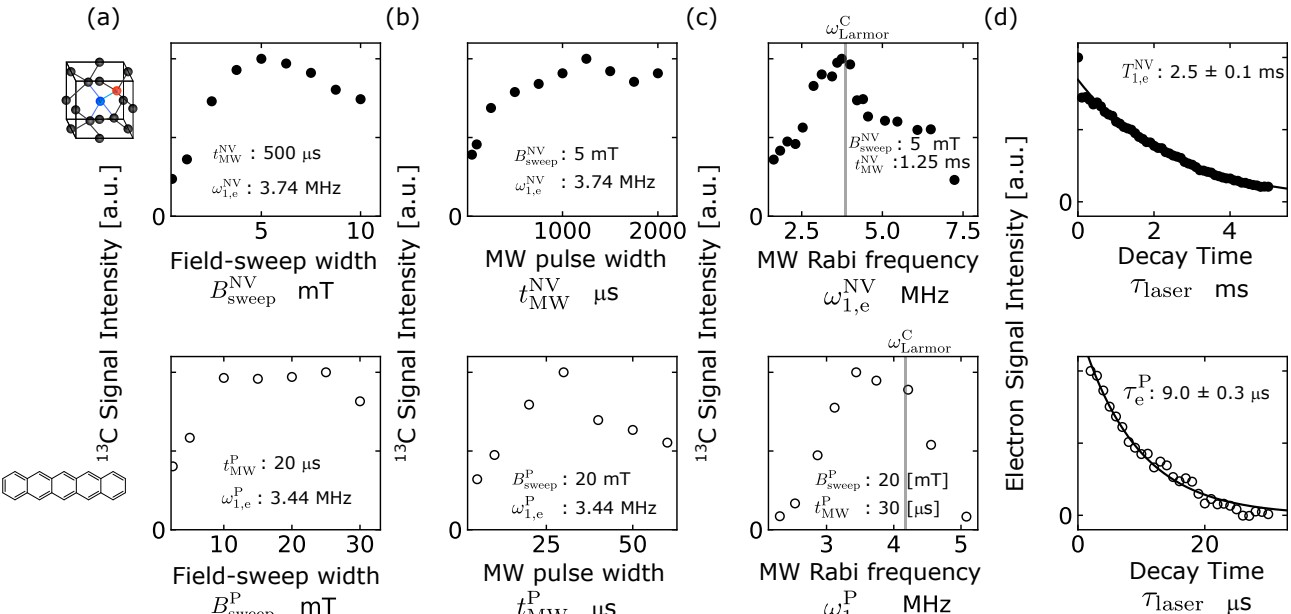

**Figure 4.** (a)-(c) Optimization of the ISE pulse sequence for microdiamonds (upper row, filled circles) and pentacene-doped benzoic acid (PBA) (lower row, open circles) through adjustment of (a) the field-sweep width $B_{\mathrm{sweep}}^{\alpha}$, (b) the microwave duration $t_{\mathrm{MW}}^{\alpha}$, and (c) the microwave intensity $\omega_{1,e}^{\alpha}$, where $\alpha = \mathrm{NV}, \mathrm{P}$ for microdiamonds and PBA, respectively. The values of the fixed parameters are indicated in each graph. The gray lines in (c) indicate the $^{13}\mathrm{C}$ Larmor frequency for the magnetic field that corresponds to the low-field horn of the EPR spectrum. (d) Longitudinal relaxation (upper graph) and lifetime decay (lower graph) of the electron spins in the triplet state of microdiamonds and pentacene, respectively.

$^{13}\mathrm{C}$ magnetization and of the EPR signal obtained in the microdiamond sample at around 0.36 T. The result that they showed similar profiles indicates that the underlying polarization mechanism is indeed ISE.

To determine the exchange probability $\xi_{\mathrm{NV}}$ appearing in Eq. (2) under the optimized ISE conditions, it is necessary to estimate the DNP-active spin-packet fraction $\eta_{\mathrm{p}}^{\mathrm{NV}}$, the triplet fraction $\eta_{t}^{\mathrm{NV}}$ of the NV$^{-}$ in the microdiamonds, and the electron polarization $\overline{P_{e}^{\mathrm{NV}}}$ averaged over the ISE sweep time. These parameters can be extracted from the experimental powder EPR spectra shown in Fig. 3 and the calculated orientational dependence of resonance magnetic field shown in Fig. 6. The optimized range of the magnetic-field sweep, indicated by the shaded areas in Fig. 3(d) and Fig. 3(f), corresponds to the region between the black lines in Fig. 6(a). From the area of this region compared to that of whole solid angle, the active spin packet fraction $\eta_{\mathrm{p}}^{\mathrm{NV}}$ was estimated to be 0.204.

Having estimated the electron polarization $P_{e}^{\mathrm{NV}}$ of the NV$^{-}$ center between the $m_S = 0$ and the $m_S = -1$ states to be 13 %, we then obtained the triplet fraction $\eta_{t}^{\mathrm{NV}} = 2/(3 - P_{e}^{\mathrm{NV}}) = 0.7$. From the measured longitudinal relaxation time $T_{1,e}^{\mathrm{NV}}$ of 2.5 ms (Fig. 4(d)), we obtained the averaged electron polarization $\overline{P_{e}^{\mathrm{NV}}}$ over the optimal pulse width $t_{\mathrm{MW}}^{\mathrm{NV}} = 1250\,\mu s$ as 0.125.



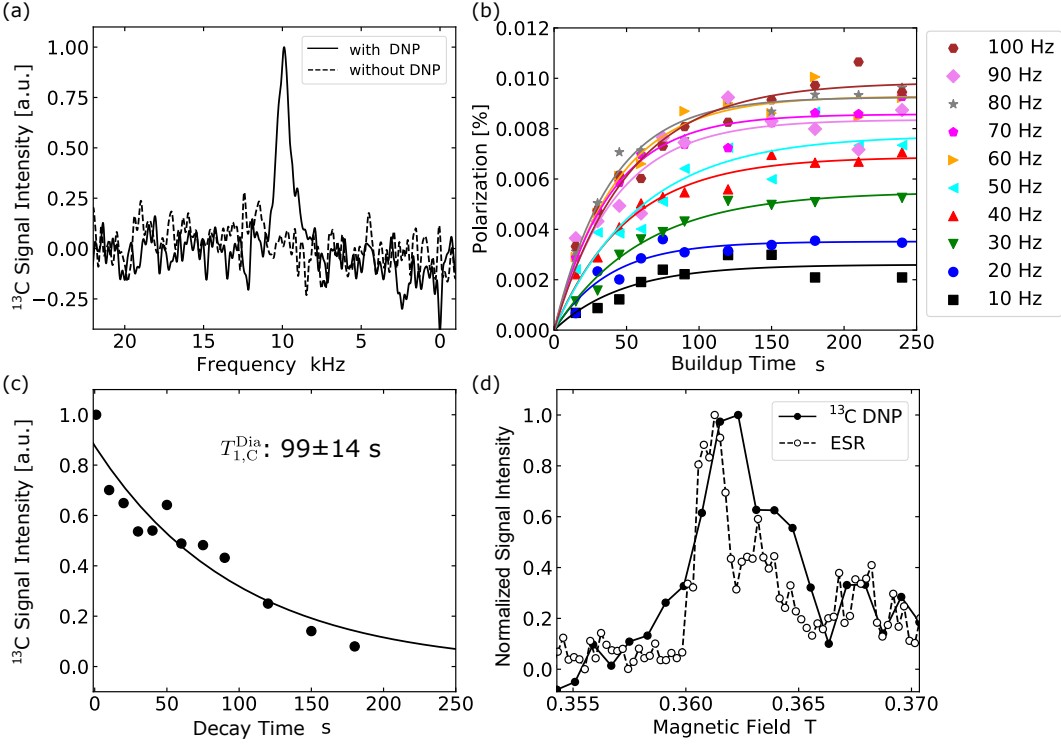

**Figure 5.** Optical hyperpolarization in microdiamonds. (a) NMR spectra of $^{13}$C in microdiamonds averaged over 4 times. The solid line shows the $^{13}$C NMR signals with DNP and the broken line shows that without DNP. The spectrum was obtained with laser repetition frequency of 100 Hz and 240 s of DNP buildup time. (b) Buildup curves of $^{13}$C polarization in microdiamonds obtained with ISE repetition rates ranging from 10 to 100 Hz. (c) The relaxation curve for the $^{13}$C NMR signal of microdiamonds. (d) The DNP and EPR field profile around 0.36 T.

In evaluating the exchange probability $\xi_{NV}$, it is important to use such experimental data that was taken under the rapid-diffusion regime that validates Eq. (4). Accordingly, the buildup curve with the ISE repetition rate of 10 Hz, which was rather low and not suitable for attaining the highest polarization, was adopted for the purpose of data fitting to Eq. (4). From the final polarization $P_{fin} = 0.0026$ %, the time constant $T_b = 47.4$ s of the buildup curve at the repetition rate $R$ of 10 Hz, the electron polarization averaged over the ISE sweep time $\overline{P_e^{NV}} \simeq 0.098$, the active spin-packet fraction $\eta_p^{NV} \simeq 0.204$, the triplet fraction $\eta_t^{NV} \simeq 0.7$, the density of triplet electrons in the microdiamonds $\rho_e^{NV} \simeq 8.9 \times 10^{17}$ cm$^{-3}$, and the $^{13}$C spin density $\rho_C^{Dia} \simeq 1.9 \times 10^{21}$ cm$^{-3}$, we obtained the exchange probability $\xi_{NV}$ to be 0.0087.

### 4.2.2 Pentacene-doped Benzoic Acid

We also successfully hyperpolarized the $^{13}$C spins in PBA through direct electron-to-$^{13}$C polarization transfer by ISE. Our results contrast with the previous works that relied on prior $^{1}$H polarization followed by $^{1}$H-$^{13}$C cross polarization (Kagawa et al., 2019). The lower part of Fig. 4(a)-(c) shows the dependence of the efficiency of dynamic $^{13}$C polarization by ISE on the



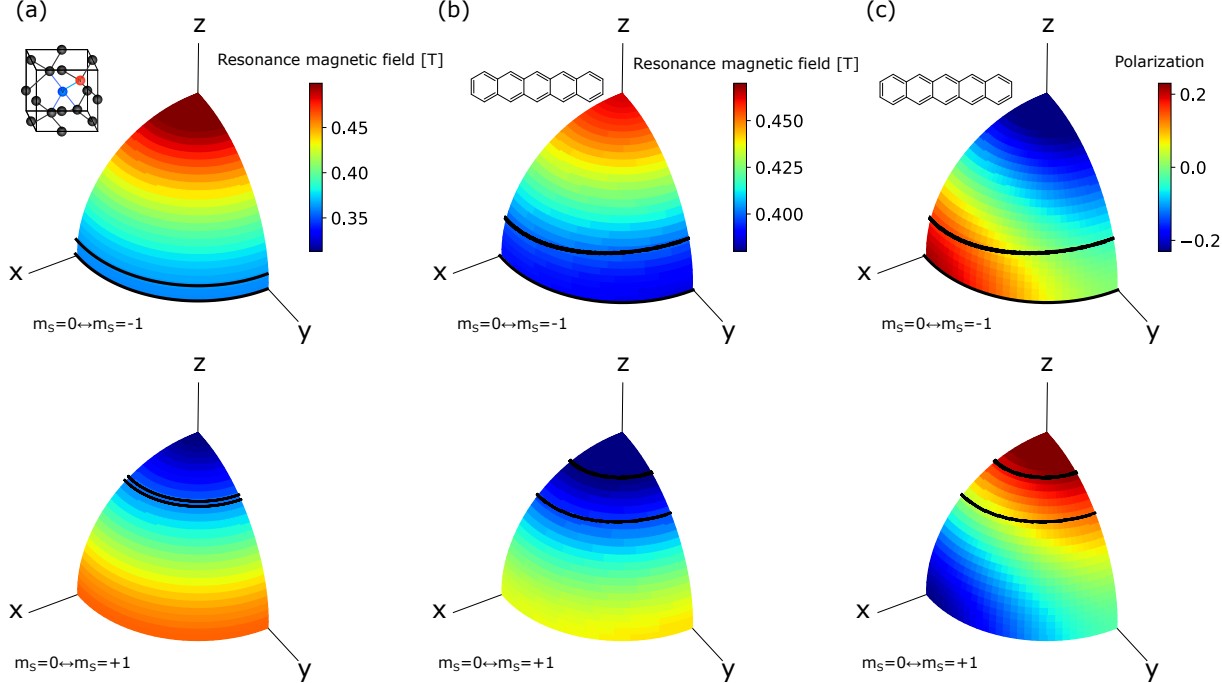

**Figure 6.** Orientational dependence of the resonant magnetic field for the electron spins in the triplet state of (a) the $\text{NV}^-$ center and (b) pentacene doped in benzoic acid at a microwave frequency of 11.66 GHz. $x$, $y$ and $z$ axes represent the principal axes of the ZFS tensor. (c) Orientational dependence of the polarization of the electron spins in the photo-excited triplet state of pentacene doped in benzoic acid in 0.39 T calculated from the zero-field population (Yu et al., 1984). The upper and lower rows correspond to the EPR transitions between the $m_S = +1$ and the $m_S = 0$ states, and between the $m_S = 0$ and the $m_S = -1$ states, respectively. The regions between the black lines correspond to the orientations for which the crystallites experience polarization transfer by ISE using the optimized field-sweep range found in the experiments demonstrated in Fig. 4.

field-sweep width $B_{\text{sweep}}^{\text{P}}$, the ISE duration $t_{\text{MW}}^{\text{P}}$, and the microwave intensity $\omega_{1,e}^{\text{P}}$, experimentally examined at 0.39 T, which corresponds to the EPR at the position where the low-field 'horn' appears in the spectrum of Fig. 3(e). The optimal conditions were found to be $B_{\text{sweep}}^{\text{P}} \simeq 20$ mT, $t_{\text{MW}}^{\text{P}} \simeq 30$ $\mu$s and $\omega_{1,e}^{\text{P}} \simeq 3.44$ MHz.

The optimal values for the field-sweep width $B_{\text{sweep}}^{\text{P}}$ and the microwave intensity $\omega_{1,e}^{\text{P}}$ differed from those $(B_{\text{sweep}}^{\text{NV}}, \omega_{1,e}^{\text{NV}})$ in the case of the microdiamond sample just by factors, whereas the optimal microwave-pulse width $t_{\text{MW}}^{\text{P}}$ was shorter than $t_{\text{MW}}^{\text{NV}}$

by more than an order of magnitude. Such striking difference can be ascribed to that the sources of polarization for PBA are the electron spins in the *metastable*, photo-excited triplet state of pentacene, while those for the diamond sample are the electron spins in the *persistent*, ground triplet state. For the PBA sample, the contact time is limited by the lifetime of the photo-excited triplet state of pentacene, whereas for the microdiamond sample it is by the electron spin-lattice relaxation time $T_{1\rho}$ in the rotating frame. As we shall discuss below, it is not those $^{13}\text{C}$ spins adjacent to the $\text{NV}^-$ centers but those located at a moderate

distance that need to receive the polarization of the electron spins of the $\text{NV}^-$ center, in order for the enhanced polarization



to be eventually transported away by $^{13}$C spin diffusion. Since the interaction between such $^{13}$C spins and the NV$^-$ centers is expected to be relatively weak, direct polarization transfer by ISE needs to be performed for a relatively longer time duration, as long as the electron magnetization is retained along the effective field in the rotating frame.

$^{13}$C NMR measurements were performed at 4.19 MHz. An enhanced $^{13}$C NMR spectrum, in comparison with that obtained
without performing ISE, is demonstrated in Fig. 7(a). Fig. 7(b) shows $^{13}$C polarization buildup behaviors for various ISE repetition frequencies. With the highest experimentally feasible ISE repetition rate $R$ of 90 Hz, the $^{13}$C polarization finally reached 0.12 %, i.e., ~3600 times the thermal $^{13}$C polarization. The $^{13}$C longitudinal relaxation time $T_{1,C}^{BA}$ was determined to be $474 \pm 30$ s (Fig. 7(c)). We estimated the active spin-packet fraction $\eta_p^{PBA}$ and the triplet fraction $\eta_t^P$ of pentacene in a similar way as in the case of the microdiamond sample. From comparison of the optimal magnetic field sweep range $B_{sweep}$ indicated
by the shaded region in Fig. 3(e) and Fig. 3(f) with the area between the black lines in Fig. 6(b)(c), $\eta_p^{PBA}$ was estimated to be 0.565. The average electron polarization $\overline{P_e^P}$ between the two triplet sublevels over the field-sweep time was estimated using the populations over the zero-field eigenstates of the triplet state of pentacene doped in benzoic acid, which is known to be $P_x : P_y : P_z = 0.44 : 0.34 : 0.22$ (Yu et al., 1984). Assuming that the external magnetic field is nearly perpendicular to the Z axis of the principal axis system of the ZFS tensor for the relevant electron-spin packets, we calculated the populations over
the triplet sublevels in the magnetic field, and obtained $P_e^P = 0.105$ and $\eta_t^P = 2/(3 - P_e^P) = 0.69$. Then, taking account of the lifetime decay with the time constant $\tau_e^P = 9$ μs (lower row of Fig. 4(d)), we determined $\overline{P_e^P}$ to be 0.03.

Using Eq. (4), we performed curve fitting of the buildup data experimentally obtained with the ISE repetition rate $R$ of 10 Hz, at which the rapid-diffusion limit is expected to be valid, and obtained $P_{fin}^{PBA}(R = 10$ Hz$) = 0.023$ % and $T_b^{PBA}(R = 10$ Hz$) = 141$ s, whence, with $\overline{P_e^P} \simeq 0.035$, $\eta_p^{PBA} \simeq 0.535$, $\eta_t^P \simeq 0.69$, $\rho_e^P \simeq 2.6 \times 10^{18}$ cm$^{-3}$, and $\rho_C^{PBA} \simeq 6.26 \times 10^{21}$ cm$^{-3}$,
we determined $\xi_{PBA}$ to be 0.035.

For the microdiamond and PBA samples, the exchange probabilities were found to be $\xi_{PBA} = 0.035$ and $\xi_{NV} = 0.0087$, respectively. To account for such a significant difference in the probability of the spin states being transferred between the electron and the $^{13}$C spins in a single shot of the ISE sequence, we again note the different types of the sources of polarization. In the case of microdiamond sample, the relevant electron spins are in the ground triplet state, persistently causing significant
local fields at the $^{13}$C sites in the vicinity. They would create the $^{13}$C spin-diffusion barriers (Wenckebach, 2016), in which the $^{13}$C spins would not be able to transport their polarization to other $^{13}$C spins via the mutual spin flip-flop process. Unfortunately, those $^{13}$C spins that are most likely to receive spin polarization from the NV$^-$ centers are inside the barrier, and thus are least likely to distribute the enhanced polarization away.

Nevertheless, the experimental result that the bulk enhancement of the $^{13}$C polarization in the microdiamonds was indeed
realized indicates the presence of those $^{13}$C spins in the sample that are capable of both receiving the polarization from the NV$^-$ center and passing it over other $^{13}$C spins through spin diffusion. Such $^{13}$C spins ought to be located at a moderate distance from the NV$^-$ center just outside the barrier. That is, if the $^{13}$C spin are too close to the NV$^-$ center, the interaction with the electron spin would overwhelm the dipolar interaction among the $^{13}$C spins, hindering the flip-flop transitions between the $^{13}$C spins. The relatively long distance between such mediating $^{13}$C spins and the NV$^-$ centers would result in the low
exchange probability $\xi_{NV}$.



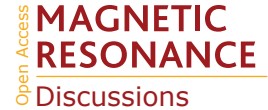

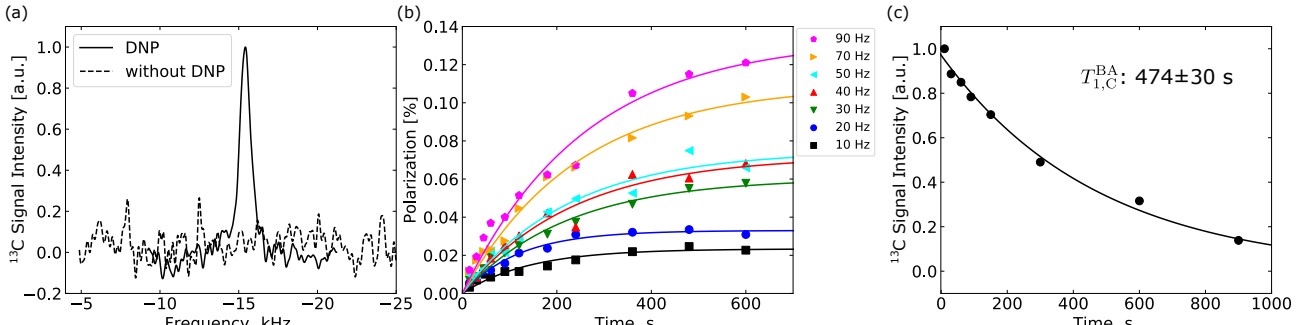

**Figure 7.** (a) $^{13}$C spectra of PBA with and without DNP. The enhanced spectrum (solid line) was obtained after performing the ISE sequence at a repetition frequency of 90 Hz for 600 s. (b) Buildup curves of $^{13}$C polarization in PBA for ISE repetition rates ranging from 10 Hz to 90 Hz. Each data point represents the $^{13}$C polarization derived from the intensity of the NMR signal accumulated over 16 times. (c) Dependence of the $^{13}$C magnetization in PBA on the time interval between DNP and NMR measurement.

Conversely, the $^{13}$C spins in PBA do not suffer from the local fields produced by the electron spins in the triplet state for most of the time during the buildup experiment, because of the transient nature of the *excited* triplet state. Indeed, the lifetime of the triplet scale, found to be ca. 9 $\mu$s (Fig. 4(d)), is three orders of magnitude shorter than the time interval between the ISE sequence even with its highest feasible repetition rate. For the $^{13}$C spins located relatively close to the pentacene molecules, the
probability of receiving the polarization from the electron spins in the triplet state is expected to be relatively high. Even if they ought to temporarily feel the strong local field and trapped inside the spin-diffusion barrier during the photo-excitation cycles, they are allowed to undergo spin diffusion for most of the time when the pentacene molecules are in the ground, diamagnetic state. Hence, the exchange probability is expected to be higher.

## 4.3 $^{13}$C spin diffusion

### 4.3.1 In diamond

In addition to the efficiency $\xi$ of direct polarization transfer from the electron spins in the triplet state to the $^{13}$C spins, spin diffusion among the $^{13}$C spins is another important factor that affects the overall enhancement of polarization. Spin diffusion is driven by the flip-flop component of homonuclear dipolar interactions, and in turn, the dipolar interaction is determined by the geometrical configuration of the relevant spins. It follows that spin diffusion is characterized by the geometrical configuration.
According to Lowe and Gade (Lowe and Gade, 1967), the $(\alpha, \beta)$-component ($\alpha, \beta = x, y, z$) of the spin-diffusion tensor $\mathcal{D}_{\alpha\beta}$ is given by

$$\mathcal{D}_{\alpha\beta} = \frac{\sqrt{\pi}}{2} \sum_i A_{0i} \alpha_{0i} \beta_{0i} \Delta_{0i}^{-\frac{1}{2}}. \tag{7}$$

Here, 0 in the subscripts represents one of the $^{13}$C sites arbitrary chosen to be the origin of the coordinate system, and the sum is taken over all $^{13}$C sites. $\alpha_{ij}(\beta_{ij})$ is the $\alpha(\beta)$ component of the internuclear vector $\mathbf{r}_{ij}$ between sites $i$ and $j$. $A_{ij}$ and $\Delta_{ij}$ are





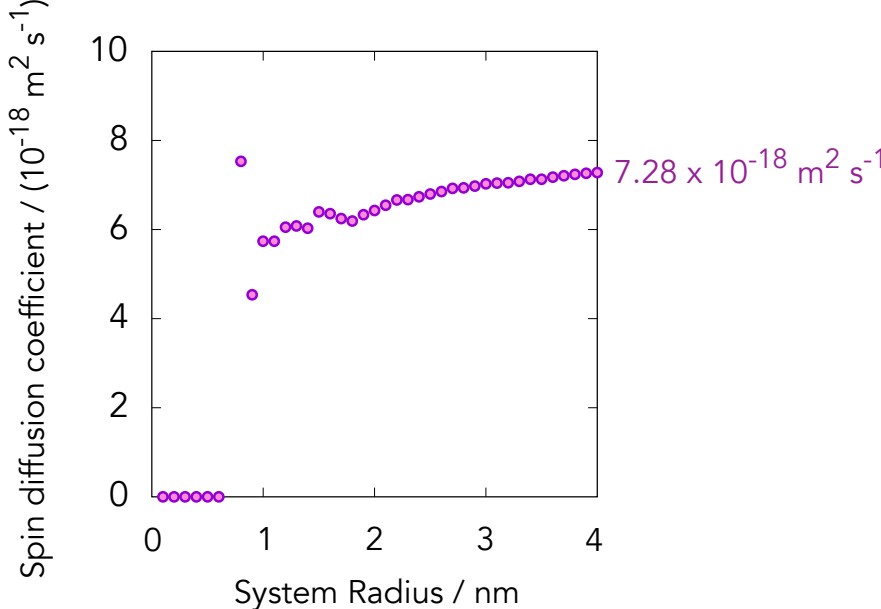

**Figure 8.** $^{13}$C spin diffusion coefficients calculated for diamond crystal using the Lowe-Gade formula (Eq. (7)) by taking account of naturally abundant $^{13}$C sites within spherical regions with various radii. 750 $^{13}$C sites are involved within the sphere with a radius of 4 nm.

expressed as

$$A_{ij} = \left(\frac{\mu_0}{4\pi}\right) \cdot \left(-\frac{1}{4}\right) \gamma^2 \hbar r_{ij}^{-3} \left(1 - 3\cos^2\theta_{ij}\right), \tag{8}$$

$$\Delta_{0i} = \frac{1}{2} \sum_{j \neq i} (B_{0j} - B_{ij})^2, \tag{9}$$

$$B_{ij} = -2A_{ij}, \tag{10}$$

where $\mu_0 = 4\pi \cdot 10^{-7}$ Hm$^{-1}$ is the vacuum permeability, $\gamma$ is the gyromagnetic ratio of $^{13}$C, and $\theta_{ij}$ is the angle between the
internuclear vector $\mathbf{r}_{ij}$ and the external static field.

     To estimate the $^{13}$C spin diffusion coefficient in the diamond sample outside the diffusion barrier where the effect of the hyperfine couplings is negligibly small, we made a table of vectors representing the carbon atomic sites in diamond. We then randomly picked up the $^{13}$C sites assuming the natural abundance of 1 %. Using the Lowe-Gade formula, we calculated the isotropic spin diffusion coefficient $D = (\mathcal{D}_{xx} + \mathcal{D}_{yy} + \mathcal{D}_{zz})/3$ by taking the $^{13}$C sites inside the spherical region with various
radii into account. As shown in Fig. 8, the calculated value of $\mathcal{D}$ increased with the size of the region that we considered, and asymptotically converged to $7.28 \times 10^{-18}$ m$^2$s$^{-1}$.





### 4.3.2 In PBA

Unlike the case of diamond where all atomic sites other than $^{13}$C are magnetically inert, estimation of the $^{13}$C spin diffusion coefficient in PBA by the Lowe-Gade formula cannot be made in a straightforward way, because of the presence of the abundant
$^1$H spins causing the considerable dipolar fields at the $^{13}$C sites. The effect of the $^1$H-$^{13}$C dipolar interaction is to lift up the degeneracy in the energy levels of the $^{13}$C spin packets, so that the flip-flop process among the $^{13}$C spins tends to not conserve the energy. Thus, the $^{13}$C diffusion rate ought to be slower than in the case if it were not for the proton spins.

In this work, we estimated the $^{13}$C spin diffusion coefficient in [carboxyl-$^{13}$C] benzoic acid from the experimental repetition-rate dependence of the initial buildup rate of the $^{13}$C polarization (Kagawa et al., 2009; Takeda, 2009). With relatively low ISE
repetition rates for which the rapid-diffusion condition is valid, the buildup rate is proportional to the repetition rate. As increasing the repetition rate, spin diffusion would no longer be able to transport the polarization completely during the time interval of ISE repetition, and the repetition-rate dependence of the buildup rate begins to saturate. The saturation of the initial buildup rate of $^{13}$C polarization was observed in the case of PBA (Fig. 9(a)). Since the initial buildup rate is independent of relaxation, one can estimate the spin diffusion rate from the data plotted in Fig. 9(a) without having to take the effect of the
former into account.

In order to reproduce the profile of the experimentally obtained initial buildup rate that increased with the repetition rate $R$ and exhibited saturation at relatively higher values of $R$, we considered a cubic region with a side length of 10.4 nm, which include one single pentacene molecule on average, and supposed that the position-dependent $^{13}$C polarization evolves in time according to the diffusion equation with a periodic boundary condition. We also assumed that the time scale of spin diffusion
is much longer than the time interval $t_{\mathrm{MW}}$ of the ISE sequence, so that the point source inside the cubic region instantly creates the $^{13}$C polarization $\overline{P_e^{\mathrm{P}}} = 0.030$ at each moment when the ISE sequence is implemented, with a probability $\xi_{\mathrm{PBA}} = 0.035$.

The profiles of numerically simulated time evolution of the net polarization for various spin diffusion coefficients $\mathcal{D}$, ranging from $8.0 \times 10^{-20}$ m$^2 \cdot$s$^{-1}$ to $1.5 \times 10^{-19}$ m$^2 \cdot$s$^{-1}$, are plotted in Fig. 9(a). We found that $\mathcal{D} = 9.75 \times 10^{-20}$ m$^2 \cdot$s$^{-1}$ is the most likely value for the $^{13}$C spin-diffusion coefficient that minimized the residual sum of squares in the case of our $^{13}$C labeled
benzoic acid sample (Fig. 9(b)).

Interestingly, the $^{13}$C spin diffusion coefficient in PBA was estimated to be smaller by two orders of magnitude smaller than that in diamond, despite the $^{13}$C enrichment in the former sample. This is ascribed to the presence of the $^1$H spins that slows down the flip-flop process between the $^{13}$C spins in PBA, and to the dense packing of the carbon atoms in the diamond crystal. From the DNP point of view, $^{13}$C spin diffusion is desirable to be as fast as possible. One way to make it is to continuously
apply $^1$H decoupling throughout the buildup experiment (Negoro et al., 2010). This strategy, however, is not practical because of the complexity of the hardware that realizes simultaneous application of radiofrequency and microwave irradiation and of the serious heating of the circuit. Deuteration of the sample can be an alternative way to make $^{13}$C spin diffusion faster in PBA.





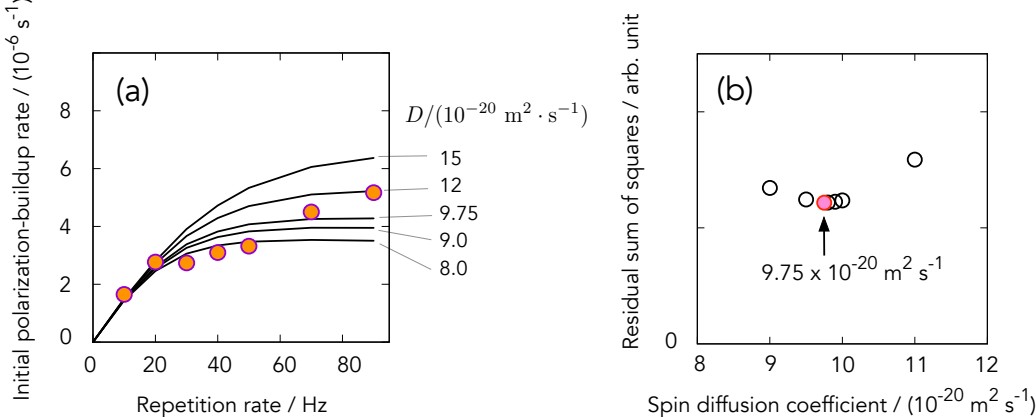

**Figure 9.** (a)Dependence of the initial buildup rate as a function of the ISE repetition rate $R$ obtained for the PBA sample. The data points indicated by the circles were obtained from the slopes of the buildup curves shown in Fig. 7(b) at time zero. Solid lines represent the $R$ dependence simulated for various spin-diffusion coefficients $\mathcal{D}$ according to the model described in the text. (b)A plot of residual sum of squares calculated for various spin diffusion coefficients, which gave the minimum for $\mathcal{D} = 9.75 \times 10^{-20}$ m$^2 \cdot$s$^{-1}$.

### 4.4 The behavior of $^{13}$C polarization buildup

It is the contribution of both the direct polarization transfer from the electrons in the triplet state to the $^{13}$C spins and $^{13}$C spin
diffusion that eventually leads to bulk $^{13}$C hyperpolarization. In reality, spin-lattice relaxation tends to drag the spin system back toward thermal equilibrium. The balance between the buildup and relaxation processes determines the profile of the $^{13}$C polarization-buildup curve, the finally-attainable bulk $^{13}$C polarization $P_{\mathrm{fin}}$, and the time required to attain $P_{\mathrm{fin}}$.

The $^{13}$C longitudinal relaxation time $T_{1,\mathrm{C}}^{\mathrm{Dia}}$ for the microdiamond sample was $99 \pm 14$ s, whereas that $T_{1,\mathrm{C}}^{\mathrm{BA}} = 474 \pm 30$ s for the PBA sample was longer than the former by a factor of ca. 5. The presence/absence of the paramagnetic electrons in
the dark state can be a factor making the difference in the relaxation time. Table 3 summarizes the exchange probability $\xi$, $^{13}$C spin diffusion coefficient $\mathcal{D}$, and the spin-lattice relaxation time $T_{1,\mathrm{C}}$. As discussed above, in PBA, the average electron polarization $\overline{P_e}$ and the $^{13}$C spin diffusion coefficient were found to be lower than those in microdiamonds. Nevertheless, the higher exchange probability $\xi_{\mathrm{PBA}}$ and longer longitudinal relaxation time $T_{1,\mathrm{C}}$ more than compensate for the lower electron polarization and the slower $^{13}$C diffusion, resulting in the higher final $^{13}$C polarization.

In order to examine the possibility that the spin polarization is leaking into the proton spin system in PBA, we implemented the ISE sequence in PBA with the parameters that we found to be the optimal for polarizing the $^{13}$C spins, and then examined whether the $^1$H magnetization was enhanced. We found that the $^1$H polarization was indeed built up to 0.16 % (not shown), higher than the final $^{13}$C polarization. Thus, deuteration of the sample, also suggested above regarding the acceleration of $^{13}$C spin diffusion, would improve the efficiency of dynamic $^{13}$C polarization using the electron spin in the photo-excited triplet
state without being mediated by the $^1$H spins.





Since only a part of the broad EPR lines is excited in triplet-DNP in the polycrystalline samples, the buildup of nuclear polarization takes place only in those crystallites that happen to be oriented in such a way that the resonant field is within the region of the adiabatic sweep. In other words, other crystallites with their resonant field being outside the region of the adiabatic sweep would not experience nuclear polarization. The size of the crystallites is such that spin diffusion across the boundaries of adjacent crystallites is negligibly slow. It follows that those inert crystallites continue to be inert, and polarization speckle would develop in the sample. In this respect, the relatively compact EPR powder spectrum of pentacene compared to the broader $NV^-$ powder spectrum is an advantage for the former, as the larger number of the crystallites can be excited. We estimate the fraction of the crystallites that contribute to the DNP process is 0.21 for microdiamonds and 0.565 for PBA. Considering this, the local $^{13}C$ polarization within the DNP-active crystallites approaches 0.05 % for microdiamonds, and 0.21 % for PBA, whereas the $^{13}C$ NMR transition in the other inert crystallites remain unpolarized.

### 4.5 Summary and prospects

We have demonstrated room temperature optical $^{13}C$ DNP using the $NV^-$ centers and the pentacene molecules. The $^{13}C$ polarization was successfully built up in microdiamonds to 0.01 %, and in PBA to 0.12 %. The difference between these polarization is due to a multitude of the favoring/disfavoring factors.

In the diamond sample, the contact time $t_{MW}$ in the ISE sequence can be set much longer than that in PBA, thanks to the persistent nature of the ground triplet state of the $NV^-$ center. However, the permanent nature of the paramagnetic $NV^-$ system turns into a disadvantage for the next, spin diffusion step; paramagnetic electrons of the $NV^-$ centers cause a significant local field around the $^{13}C$ spins in the vicinity, creating the spin diffusion barriers around them. Unfortunately, those $^{13}C$ spins that are most likely to be directly polarized are inside the barrier, so that they are least likely to distribute the polarization away outside the barrier. The $^{13}C$ spins just outside the barrier can both receive the electron polarization and undergo spin diffusion. However, their relatively long distance to the $NV^-$ center leads to the low exchange probability $\xi^{NV} \simeq 0.0087$. Moreover, many other paramagnetic electron defects in diamond, such as P1 centers, play the same detrimental role for an efficient spread of the nuclear polarization.

The PBA system is free from the problem of diffusion barrier due to the transient nature of the paramagnetic electrons in the triplet state, which decays to the 'dark', diamagnetic, ground state within the lifetime of the order of several microseconds. Even though the contact time is limited by the rather short lifetime, the efficiency of the direct, electron-to-$^{13}C$ polarization transfer was found to be higher than that for the case of the microdiamond sample.

However, the effect of abundant $^1H$ spins reduces both the efficiency of the direct polarization transfer and of $^{13}C$ spin diffusion. Even though one carefully optimizes the microwave amplitude in favor of the $^{13}C$ spins, a considerable amount of electron polarization was found to leak to the $^1H$ spins.

In addition, $^1H$-$^{13}C$ dipolar broadening (see SI) lifts up the degeneracy in the energy levels of the $^{13}C$ spin packets, reducing the $^{13}C$ spin-diffusion rate and thereby the buildup of overall $^{13}C$ polarization. The $^{13}C$ spin diffusion coefficient in PBA evaluated from the $R$ dependence of the initial buildup rate in comparison with numerical simulation was found to be two orders of magnitude lower than the spin-diffusion coefficient for the naturally abundant $^{13}C$ spins in diamond.





The optical DNP experiments in NV⁻-containing diamonds reported so far used all continuous light irradiation, whereas we have demonstrated a pulsed optical excitation for the first time. The polarization of the electrons in the ground triplet state created through the pulsed photo-excitation can be lower than that obtained through continuous photo-excitation, in particular when the pulse length is shorter than the decay time of the meta-stable singlet state $^1$E to the triplet ground state $^3$A$_2$, which is ca. 300-400 ns (Doherty et al., 2013) (see Fig. 1). On the other hand, the pulsed excitation can be beneficial in preventing

NV⁻ from photo-ionization to NV$^0$ ($S = 1/2$) (Loretz et al., 2017; Chen et al., 2017).

    How could we improve optical DNP experiments, with these results in mind? Ajoy et al. showed a smart way to strongly increase the number of excited crystallites, by choosing a low magnetic field, where the total broadening is $< 2D$ (a range, where the Zeeman term is smaller than the ZFS term), where a $^{13}$C polarization of 0.25 % (detected in a high-field NMR spectrometer) was achieved (Ajoy et al., 2018a). A direct comparison of the results is practically impossible, due to differences

in diamond samples (varying concentrations in $^{13}$C, NV⁻ and P1 centers) (Parker et al., 2019). It would be interesting to repeat DNP experiments of both samples at this low-field setup. At higher fields, a frequency-swept ISE experiment (Can et al., 2017) (in contrast to our field-swept version) making use of ultra-wide bandwidth MW chirp pulses (Segawa et al., 2015) could increase the number of excited crystallites. From the diamond engineering side, highly pure crystals with a low concentration of paramagnetic defects (P1 centers and others) would be desirable, extending both, electron and nuclear spin

relaxation times. Diamonds fabricated using chemical vapour deposition (CVD) may fulfill this criterion, albeit for a very high price. We could not perform DNP experiments on our 100 nm nanodiamonds, since the sample amount was too low. The best DNP enhancement for 100 nm nanodiamonds reported in the literature is only $\epsilon = 3$ (compared with the thermal signal intensity at 7 T) (Ajoy et al., 2020). The difficulty in achieving DNP in real nano-particles is directly linked to the drastic reduction of nuclear relaxation times $T_{1,C}$, finally arriving at values shorter than 1 s for 5 nm nanodiamonds (Casabianca

et al., 2011). Methods to eliminate paramagnetic surface defects, known as dangling bonds, as well as ways to transfer the polarization outside the diamond crystals, are yet to be investigated.

    In this respect, pentacene-doped organic host crystals can highly compete; $^1$H spins in 150 nm size crystals of pentacene in $p$-terphenyl were optically polarized at room temperature in water, achieving a polarization of 0.086 % (Nishimura et al., 2019). Optically hyperpolarized benzoic acid crystals doped with pentacene were dissolved and a DNP enhanced liquid-state $^1$H NMR

signal of benzoic acid was measured (Negoro et al., 2018). These examples show impressive advantages on the material side of organic crystals with a controlled pentacene doping. However, organic crystals suffer from melting or damage when they are irradiated with high power laser beams, whereas this is not a big problem for diamonds. The quest for new defects with ODMR signals, which has just started, will also lead to a variety of new candidates for optical triplet DNP (Segawa and Shames, 2020). Hopefully, our comparative approach serves as a guideline to select or even engineer improved systems for room-temperature

hyperpolarization of dilute/low-$\gamma$ spins.

*Data availability.* Experimental data are available upon request from the corresponding authors.





**Table 3.** Comparison of parameters of the DNP experiment.

| Sample | $^{13}$C in Microdiamond | $^{13}$C in PBA |
|---|---|---|
| $P_{\text{fin}}$ [%] | 0.01 | 0.12 |
| $T_{1,\text{C}}$ [min] | 1.65 | 7.9 |
| $\xi$ [%] | 0.87 | 3.5 |
| $\eta_{\text{p}}$ | 0.20 | 0.54 |
| $\eta_{\text{t}}$ | 0.70 | 0.69 |
| $\overline{P_e}$ [%] | 9.8 | 3.0 |
| $\rho_n/\rho_e$ | 2200 | 2500 |
| $R$ [Hz] | 100 | 90 |
| $\mathcal{D}$ [m$^2$s$^{-1}$] | $7.28 \times 10^{-18}$ | $9.75 \times 10^{-20}$ |

*Author contributions.* K. Miyanishi, T.F.S. and M.N. designed the experiment and K. Miyanishi carried out the experiments under the supervision of T.F.S., A.K., M.K. and M.N.. S.O., T.O., H.T., S.T., F.T.-K.S., D.T., R.I. and M.S. prepared nanodiamonds containing NV$^-$ centers. I.O., H.A. and N.M.prepared microdiamonds containing NV$^-$ centers. Y.W. and K.T. performed simulations estimating diffusion coefficients. A.I.S. estimated the electron spin concentrations of the nanodiamonds from CW EPR spectra. K. Morita estimated the P1 center concentration of the microdiamonds under the supervision of I.O. and N.M.. K. Miyanishi, T.F.S., K.T. and M.N. analysed and interpreted the data and K. Miyanishi, T.F.S., K.T. and M.N. wrote the manuscript with input from all authors.


*Competing interests.* The authors declare that they have no conflict of interest.

*Acknowledgements.* This work is supported by MEXT Quantum Leap Flagship Program (MEXT Q-LEAP) Grant Number JPMXS0120330644,
JPMXS0118067634 and JPMXS0118067395, PRESTO (JST grant number JPMJPR1666, JPMJPR18G5, and JPMJPR18G1), and Osaka University's International Joint Research Promotion Program. T.F.S. acknowledges The Branco Weiss Fellowship–Society in Science, administered by the ETH Zurich and Prof. Shirakawa for hosting him as a Guest Research Associate at Kyoto University. K. Miyanishi is supported by JSPS KAKENHI No. 19J10976 and Program for Leading Graduate Schools: Interactive Materials Science Cadet Program. K.T. is supported by JST CREST (Grant No. JPMJCR1873).





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
