# Peer review of "Room temperature hyperpolarization of polycrystalline samples with optically polarized triplet electrons: Pentacene or Nitrogen-Vacancy center in diamond?"

_Magnetic Resonance, 2020_

## Referee Comment (RC1) · Jeffrey Reimer (Referee) · 3 Jan 2021

The manuscript by Miyanishi et al describes phenomenology, and detailed analysis, of 13C polarization enhancement by DNP in two systems: diamond and benzoic acid. The authors create athermal electron polarizations in electron triplet states via optical pumping, then drive that polarization to 13C nuclei via the "integrated solid effect," which in these cases means matching electron Rabi frequencies to nuclear Larmor frequencies. That the authors achieve modest polarizations on the 13C reservoir is not surprising; this manuscript is particularly noteworthy in that the both the electron and

nuclear resonances are interrogated and that the detailed mechanism is revealed by a detailed model. Readers that are interested in the "physics" of DNP in an accessible and thoughtful paper need look no further that this work.

For the authors consideration I include some comments below, though I am recommending the manuscript be published as is.

Lines 19-33: Often neglected in modern times is the extensive DNP work in "inorganic" semiconductors such as GaAs (conflict of interest: I am an author on some of these works). Deep conduction band photoexcited electrons, or photoexcited electrons captured at various defects, afford a surprising array of spin physics...I call attention to any of the papers by Meriles et. al. at CCNY.

Lines 54-56: Although later in the paper you press the advantage of photoexcited states for DNP, it might be worth emphasizing that point here: once the OP-DNP is accomplished, the effects of unpaired electron spins vanish and the full suite of high resolution solids NMR in diamagnetic materials becomes available.

Line 65: It is kind of the authors to call attention to the 2010 paper by King et al; though I note the mechanism by which nuclear hyperpolarization occurs in those high-filed pumping experiments has not been ascertained.

Section 2.1 is particularly well written and accessible.

Section 4.1 and Figure 3: These EPR results are particularly compelling in making the case for the ISE mechanism.

Lines 162-163: I believe the shuttling experiments from Ajoy et al may be the first optically polarized and 13C hyperpolarized measurements, albeit by shuttling the samples into a high field NMR system.

Line 199: It is interesting to compare the 13C diamond T1 values ($\sim$100 seconds) with those reported in https://doi.org/10.1038/s41467-019-13042-3. If I red the graphs correctly, the samples used in Ajoy et al have a T1 value of $\sim$50 seconds at that same

field strength, consistent with the higher P1 spin density is the Ajoy samples.

---

## Short Comment (SC1) · 5 Jan 2021

Since I am interested in DNP, but mostly working in EPR, I have a few questions and comments:

1) Section 3.2. mentions a cavity. Could you give some details, i.e. the bandwidth? Would frequency sweeps be possible in principle? What is the maximum electron Rabi frequency you can achieve?

2) Section 4.1. The measurement in Figure 3 (b) (optically polarized NV Powder spectrum) took 7h. I have no experience at all with these systems and setups, but I was wondering why it takes so long?

3) Figure 3 (f) I do not think it is absolutely necessary, but since you already use EasySpin to simulate the triplet spectra, it would be straight-forward to include the non-equlibrium populations (either via "Exp.Temperature" in EasySpin version 5 or via "Sys.Pop" in version 6 (developer version)).

---

## Referee Comment (RC2) · Anonymous Referee #2 · 9 Jan 2021

The authors use optically polarized electron triplets in combination with integrated solid-effect DNP to polarize 13C nuclei in NV- doped diamond and also in pentacene doped benzoic acid. The authors extract spin exchange probabilities from the EPR and NMR data and calculate 13C spin diffusion coefficients for each sample in order to assess the suitability of each system for generating bulk 13C polarization in the host matrix. The exchange probability is extracted by modeling the build-up of 13C polarization in the rapid-diffusion limit and spin diffusion coefficients are calculated from the Lowe-Gade formula for the NV case and estimated based on previous works for the

pentacene case. The authors suggest that the triplet state lifetime and the presence of heteronuclear dipolar coupling accounts for the difference in observed exchange probabilities for the two systems.

This work calls attention to several parameters (source lifetime, T1n, spin-spin coupling effect) that should be considered when choosing a polarized electron triplet source and matrix to maximize the bulk polarization of a particular nucleus in a sample, specifically low gamma. Most interestingly, this work shows how the exchange probability can be extracted from NMR/EPR data and presents a metric to assess the general suitability of optically hyperpolarized systems for efficient nuclear polarization of the surrounding matrix.

The paper is well-written and of interest to the MR community though it is lacking in some areas with respect to proper references and literature knowledge.

Specific comments and also technical/reference comments:

1. Line 22. The optically generated spin polarization does depend on the magnetic field in specific instances when one considers level crossings and spin mixing effects.

2. Line 25. Triplet-DNP was not originally demonstrated in a single crystal of naptha-lene with pentacene, there is at least one earlier reference to the case where Kesteren, Wenckebach and Schmidt acheived this in a different system (Fluorene doped with deuterated phenanthrene)[https://doi.org/10.1103/PhysRevLett.55.1642].

3. Line 45. Needs a reference for order of singlet states [https://doi.org/10.1103/PhysRevB.98.085207]

4. Line 59. I'm repeating J. Reimer's comment here, this is not DNP, no microwaves were used in this experiment by King 2010.

5. Line 66. As J. Reimer said, Ajoy paper is first example of ISE used in NV diamond, though detection done at higher field, and was frequency swept not field swept.

6. Paragraph starting at line 74 provides context that is not relevant to the study, I would suggest removing it.

7. Line 99. The authors should be more specific about the rapid diffusion model when it is first introduced, under what specific conditions is it valid?

8. When extracting spin diffusion coefficients the authors do not include the effect of paramagnetic sinks, could the authors clarify this point?

9. Line 163: The authors state: "we estimated the initial electron polarization PNVe of the NV− center between the mS = 0 and the mS = −1 states right after laser irradiation to be 13 % from the ratio of the area intensity of the EPR line of the NV− centers integrated over the field-sweep range to the area intensity of the EPR lines of all S = 1/2 electron spins (177 ppm)." I am not sure this is entirely valid unless one includes the effects of optical density and scatter. The entire volume of S=1/2 electron spins is measured in the EPR cavity, however wouldn't only a fraction of NV- spins be excited based on scattering, optical density and the laser spot size of the sample? Could the authors clarify this point?

10. From what I understand, the 13% value is extracted from the data shown in 3b, for the nanodiamonds. Why is this same value applied to the microdiamonds as opposed to using the data from 3c and known P1 concentrations in the microdiamonds? I would also expect the penetration depth to be quite different for the microdiamonds.

11. Line 385, Though this is the first report I've seen to haev used a ns pulsed laser source, there are many other cases in which laser schemes are used with gating for the laser + RF + microwaves to polarize a small number of neighbouring 13C nuclei. (e.g. https://doi.org/10.1103/PhysRevLett.120.060405). I would suggest changing the 'continuous light irradiation' to continuous wave laser light sources to be more clear.

12. Line 405. There have been a number of studies focused on methods to eliminate surface defects in diamond as this can be lead to very short coherence lifetimes for

NV centers near the surface, this has included different baking/oxidizing techniques as well as different surface coating methods. [e.g. a few ref: Analytical and Bioanalytical Chemistry volume 407, pages7521–7536(2015), and also Nano Lett. 2013, 13, 10, 4733–4738 ] In addition, there have been a number of proposed schemes for transferring polarization from the diamond matrix to external spins, though none have been applied successfully to a larger volume. [Nano Lett. 2014, 14, 5, 2471–2478; Nano Lett. 2018, 18, 3, 1882–1887 ]

13. Line 413.The quest to find new defects in semiconductors that can be optically polarized has been ongoing for at least a decade, though it is found more so in the quantum information processing literature (Nature Physics volume 3, pages153–159(2007).

14. The authors may find the following (very) recently published work interesting: "Scaling analyses for hyperpolarization transfer across a spin-diffusion barrier and into bulk solid media" https://doi.org/10.1039/D0CP03195J

Comments on supplementary:

1. I would suggest that the authors add a reference to the following paper [E. Rej et al. J. Am. Chem. Soc. 2017, 139, 1, 193–199 ] (in which the ESR of dangling bonds in nanodiamonds are discussed and shows the g-factors and where ESR lineshapes of P1 and surface radicals overlap). 2. Line 9: there are more than three lines here, if the authors include all the features of dangling bonds, one is dealing with multiple overlapping resonances (E. Rej: J. Am. Chem. Soc. 2017, 139, 1, 193–199). 3. In the supplementary text the authors report T1,C in microdiamonds to be 99 +/- 14 s but then in figure 3(c) the authors show T1,C to be 85+/- 5 s. Please clarify this point.

---

## Short Comment (SC2) · 10 Jan 2021

Dear Authors,

I very much enjoyed reading your work. In this context and it led to two questions regarding the resonance condition that you have employed.

First, in your work you appear to have chosen to achieve resonance for perfectly aligned electron spins. Misalignment then results in a loss of resonance and interferes with the polarisation process.

[Figure]

In Sec V of Q. Chen, I. Schwarz, F. Jelezko, A. Retzker and M.B. Plenio. Optical hyperpolarization of 13C nuclear spins in nanodiamond ensembles. Phys. Rev. B 92, 184420 (2015) we have also examined an alternative resonance condition chosen for NV orientations that are orthogonal on the external magentic field. Theory suggests that this choice leads to a higher robustness to misorientations as a larger fraction of NVs will be found within a certain range of rotation angles. This was tried out experimentally in J. Scheuer et al. Optically induced dynamic nuclear spin polarization in diamond. New J. Phys. 18, 013040 (2016) but no systematic comparison was done due to the limitation of that particular experiment.

It would be interesting to see whether your eperimental setup may allow for a more careful assessment of the merits of this choice of resonance condition.

Second, in the same work Q. Chen, I. Schwarz, F. Jelezko, A. Retzker and M.B. Plenio. Optical hyperpolarization of 13C nuclear spins in nanodiamond ensembles. Phys. Rev. B 92, 184420 (2015) we have compared a driving scheme in the |0> <-> |m=+1> transition with a scheme that uses a double quantum transition |m=-1> <-> |0> <-> |m=+1> which exhibits a significantly enhanced robustness to misalignments of the NV center. This benefits of this approach have not been tested yet experimentally and it would seem interesting to do so in your current experiment.

Yours sincerely Martin Plenio
* * *

---

## Author Comment (AC1) · 21 Jan 2021

Author comment for "mr-2020-36"

*Room temperature hyperpolarization of polycrystalline samples with optically polarized triplet electrons: Pentacene or Nitrogen-Vacancy center in diamond?*

We thank the two reviewers and the two authors of the "short comments" for their relevant and valuable comments, questions and suggestions. Please find below our responses including revised passages of our manuscript.

**mr-2020-36-RC1**

1. *Readers that are interested in the «physics» of DNP in an accessible and thoughtful paper need look no further that this work. For the authors consideration I include some comments below, though I am recommending the manuscript be published as is.*

We thank the reviewer #1 for positively evaluating our work and recommending publication.

2. *Lines 19-33: Often neglected in modern times is the extensive DNP work in "inorganic" semiconductors such as GaAs (conflict of interest: I am an author on some of these works). Deep conduction band photoexcited electrons, or photoexcited electrons captured at various defects, afford a surprising array of spin physics…I call attention to any of the papers by Meriles et. al. at CCNY.*

We realize that a number of previous amazing works, including optical pumping in semiconductors as the reviewer pointed out, form a vast basis that we could stand on to have worked along the current topic. Since we do not intend to comprehensively review various branches of nuclear polarization using light, and since such a work (a very instructive one) has recently be published by Dieter Suter in Magnetic Resonance (Magn. Reson., 1, 115–139, 2020, https://doi.org/10.5194/mr-1-115-2020), we would like just to mention and cite the most relevant, early papers on optical nuclear polarization in molecular crystals.

3. *Lines 54-56: Although later in the paper you press the advantage of photoexcited states for DNP, it might be worth emphasizing that point here: once the OP-DNP is accomplished, the effects of unpaired electron spins vanish and the full suite of high resolution solids NMR in diamagnetic materials becomes available.*

The context at this position (lines 54-56) deals with both pentacene and NV$^-$, and the advantage of the vanishing electron paramagnetism is relevant only for the former case.

4. *Line 65: It is kind of the authors to call attention to the 2010 paper by King et al; though I note the mechanism by which nuclear hyperpolarization occurs in those high-filed pumping experiments has not been ascertained.*

We agree that we should not call such an «all-optical» (i.e., «microwave-free») hyperpolarization process «DNP» (see also similar comment #5 by reviewer #2). Motivated by this comment (and also

comments #5/#6 of reviewer 2), we have revised the corresponding paragraph (lines 59-67 in first version of the manuscript) with the goal to describe the polarization mechanisms more precisely:

"Microwave-free optical hyperpolarization for a bulk ensemble of $^{13}$C nuclear spins using NV$^-$ centers in a diamond single crystal was first demonstrated at cryogenic temperature in a magnetic field of 9.4 T (King et al., 2010). Optically-pumped electron spins were used for nuclear hyperpolarization by exploiting an excited-state level anticrossing at 50 mT, followed by sample shuttling to a magnetic field of 4.7 T for NMR detection (Fischer et al., 2013). The method was generalized to a broader range of magnetic fields and correspondingly different orientations of the NV$^-$ center by adding microwave irradiation (Álvarez et al., 2015). For a single crystal of diamond, the $^{13}$C polarization of 6% at room temperature has been achieved via DNP as a combination of the thermal mixing and the solid effects (King et al., 2015). Recently, DNP using the frequency-swept ISE on NV$^-$ in powdered microdiamonds in a magnetic field of as low as ca. 30 mT has been reported by Ajoy et al., who took advantage of the reduced width of the anisotropic electron paramagnetic resonance (EPR) powder pattern of the NV$^-$ centers (Ajoy et al., 2018a,b)."

5. Section 2.1 is particularly well written and accessible. Section 4.1 and Figure 3: These EPR results are particularly compelling in making the case for the ISE mechanism.

We appreciate these comments.

6. Lines 162-163: I believe the shuttling experiments from Ajoy et al may be the first optically polarized and $^{13}$C hyperpolarized measurements, albeit by shuttling the samples into a high field NMR system.

We agree that Ajoy et al. (DOI: 10.1073/pnas.1807125115) show in Fig. 4A an NV$^-$ EPR powder spectrum. However, this is not detected in an EPR spectrometer, but represents the $^{13}$C polarization, as function of the MW frequency.

7. Line 199: It is interesting to compare the 13C diamond T1 values (ca. 100 seconds) with those reported in https://doi.org/10.1038/s41467-019-13042-3. If I red the graphs correctly, the samples used in Ajoy et al have a T1 value of ca. 50 seconds at that same field strength, consistent with the higher P1 spin density is the Ajoy samples.

Thank you for the comment. We have studied Fig. 4a of Ajoy et al. 2019 plotting the magnetic field dependent $T_1$ relaxation (https://doi.org/10.1038/s41467-019-13042-3). Sample 3 with size of 200 um and P1 concentration of 200 ppm is the closest to our microdiamonds. However, we are not sure about the true values, since the y-axis has a unit $1/T_1$ (mHz), which would lead to unphysically long relaxation times.

**mr-2020-36-RC2**

The paper is well-written and of interest to the MR community though it is lacking in some areas with respect to proper references and literature knowledge.

We thank reviewer #2 for both encouraging and complementing knowledge based on the reviewer's expertise.

1. Line 22. The optically generated spin polarization does depend on the magnetic field in specific instances when one considers level crossings and spin mixing effects.

The reviewer's comment makes sense in this introduction part, where we should describe a general background of the topic. Accordingly, we revised the sentence to mention this point with citations to the papers by 1993 He et al. «NV center – Level anticrossing in the $^3A$ ground state» (https://doi.org/10.1103/PhysRevB.47.8809), 1994 Corval et al. «Resonant Intersystem Crossing in Pentacene» (https://doi.org/10.1021/j100081a024) and 2018 Sosnovsky and Ivanov «Magnetic field dependence of triplet-state ONP: theoretical analysis in terms of level anti-crossings» (https://doi.org/10.1080/00268976.2018.1504996):

in line 22, we added, "…, except that the magnetic field happens to be such that level (anti-)crossings take place (He et al., 1993; Corval et al., 1994; Sosnovsky and Ivanov, 2019)."

Also, we changed the sentence in l. 52 by removing its second part "…and is independent of temperature and magnetic field.":

Moreover, the triplet population depends on the host molecule.

2. Line 25. Triplet-DNP was not originally demonstrated in a single crystal of naphthalene with pentacene, there is at least one earlier reference to the case where Kesteren, Wenckebach and Schmidt acheived this in a different system (Fluorene doped with deuterated phenanthrene)[https://doi.org/10.1103/PhysRevLett.55.1642].

Response: We thank the reviewer for pointing out this. Indeed, the first example of triplet-DNP was reported in *p*-dibromobenzene doped with *p*-dichlorobenzene (doi: 10.1016/0022-2364(80)90128-6), and then in fluorene doped with phenanthrene (doi: 10.1016/0009-2614(82)83344-7). In the revised manuscript, we mentioned these works with citations.

3. Line 45. Needs a reference for order of singlet states [https://doi.org/10.1103/PhysRevB.98.085207]

We have added this reference, which theoretically derives the order of the singlet states, when first mentioning Fig. 1 in the revised manuscript.

The electronic structure for these systems are shown in Fig. 1 (Takeda, 2009; Doherty et al., 2013; Rogers et al., 2008; Acosta et al., 2010; Thiering and Gali, 2018).

4. Line 59. I'm repeating J. Reimer's comment here, this is not DNP, no microwaves were used in this experiment by King 2010.

See answer to comment #4 of reviewer #1.

5. Line 66. As J. Reimer said, Ajoy paper is first example of ISE used in NV diamond, though detection done at higher field, and was frequency swept not field swept.

See answer to comment #4 of reviewer #1.

6. Paragraph starting at line 74 provides context that is not relevant to the study, I would suggest removing it.

We believe that this short excursion to ODMR of single spins with the same systems (pentancene and NV- centers in diamond) can be of interest to the reader.

7. Line 99. The authors should be more specific about the rapid diffusion model when it is first introduced, under what specific conditions is it valid?

The condition for the rapid-diffusion limit is stated in line 99-100; the polarization is smoothed out within the interval of the repeated ISE sequences. We revised the sentence to make this point clearer.

We have modified the sentence in the following way: Conversely, when $^{13}$C spin diffusion is fast such that the $^{13}$C polarization is smoothed out within the interval of the repeated ISE sequences,…

8. When extracting spin diffusion coefficients the authors do not include the effect of paramagnetic sinks, could the authors clarify this point?

For the experimental determination of the spin diffusion coefficient in PBA, we tried to extract the diffusion coefficients from the initial buildup rate, i.e., the slope of the buildup curve at $t = 0$, which is independent of the relaxation effect. We have discussed this point around Eq. (6) in the original manuscript.

The estimation of the spin diffusion coefficient in diamond using the Lowe-Gade formula does not include paramagnetic sinks. The diffusion coefficient should be reasonable for those $^{13}$C spins located outside the diffusion barrier.

9. Line 163: The authors state: "we estimated the initial electron polarization PNVe of the NV-center between the mS = 0 and the mS = - 1 states right after laser irradiation to be 13 % from the ratio of the area intensity of the EPR line of the NV- centers integrated over the field-sweep range to the area intensity of the EPR lines of all S = 1/2 electron spins (177 ppm)." I am not sure this is entirely valid unless one includes the effects of optical density and scatter. The entire volume of S=1/2 electron spins is measured in the EPR cavity, however wouldn't only a fraction of NV- spins be excited based on scattering, optical density and the laser spot size of the sample? Could the authors clarify this point?

We thank the reviewer for this comment. Indeed, we agree that optical density and scattering are particle size dependent effects. Moreover, our initial estimation of the optically-induced electron spin polarization in the NV-nanodiamonds suffers from the fact that the calibration is performed as a pulse experiment, where relaxation effects are unavoidable.

Therefore, we will replace this estimation in NV-nanodiamonds by a direct estimation of the signal enhancement in the NV-microdiamonds. By accumulating a field-swept ESE signal of the microdiamonds without laser irradiation over night, we obtained an enhancement by optical polarization of ca. 170. Multiplying this with the thermal electron spin polarization of 0.095 %, we obtained an enhanced polarization of 16 % (previously estimated 13 % for nanodiamonds). Taking the noise in the "dark" spectrum as the main source of uncertainty, we obtain an error of +/- 20%. This data will be replaced in Fig. S1 of the SI. Dependent parameters in the main text are corrected.

With this direct estimation of the optical electron polarization in the microdiamonds, we can avoid all uncertainties that were correctly addressed by reviewer #2. The new SI figure, comparing the two spectra with and without light, is shown below.

[Figure]

10. From what I understand, the 13 % value is extracted from the data shown in 3b, for the nanodiamonds. Why is this same value applied to the microdiamonds as opposed to using the data from 3c and known P1 concentrations in the microdiamonds? I would also expect the penetration depth to be quite different for the microdiamonds.

See answer to comment #9.

11. Line 385, Though this is the first report I've seen to haev used a ns pulsed laser source, there are many other cases in which laser schemes are used with gating for the laser + RF + microwaves to polarize a small number of neighbouring 13C nuclei. (e.g. https://doi.org/10.1103/PhysRevLett.120.060405). I would suggest changing the 'continuous light irradiation' to continuous wave laser light sources to be more clear.

Indeed, in the ODMR spectroscopy of single NV centers, ns pulsed or modulated lasers are widely used. To be more precise, we have modified the sentence in the following way:

The optical bulk DNP experiments in NV$^{-}$-containing diamonds reported so far used all continuous-wave laser light sources, whereas we have demonstrated a pulsed optical excitation for the first time.

12. Line 405. There have been a number of studies focused on methods to eliminate surface defects in diamond as this can be lead to very short coherence lifetimes for NV centers near the surface, this has included different baking/oxidizing techniques as well as different surface coating methods. [e.g. a few ref: Analytical and Bioanalytical Chemistry volume 407, pages7521 - 7536(2015), and also Nano Lett. 2013, 13, 10, 4733 - 4738 ] In addition, there have been a number of proposed schemes for transferring polarization from the diamond matrix to external spins, though none have been applied successfully to a larger volume. [Nano Lett. 2014, 14, 5, 2471 - 2478; Nano Lett. 2018, 18, 3, 1882 - 1887 ]

We agree that the "surface control" of (nano-)diamonds is an active field of research in the NV community. However, we feel that regarding reduction of surface (or subsurface) radicals, a breakthrough is still missing. We have slightly reformulated the sentence and add the proposal by 2014 Abrams et al. as a citation for polarization of external spins.

Methods to eliminate paramagnetic surface defects, known as dangling bonds, as well as ways to transfer the polarization outside the diamond crystals (Abrams et al., 2014), are yet to be experimentally demonstrated.

13. Line 413.The quest to find new defects in semiconductors that can be optically polarized has been ongoing for at least a decade, though it is found more so in the quantum information processing literature (Nature Physics volume 3, pages153 - 159(2007).

We thank the reviewer for this comment. We have added the following sentence in the revised manuscript:

The field of semiconductor spintronics presents a rich source of inspiration (Awschalom and Flatté, 2007).

14. The authors may find the following (very) recently published work interesting: "Scaling analyses for hyperpolarization transfer across a spin-diffusion barrier and into bulk solid media" https://doi.org/10.1039/D0CP03195J

We thank the reviewer for making us aware of this new theoretical approach in describing nuclear spin diffusion in the presence of spin-diffusion barriers.

Comments on supplementary:

1. I would suggest that the authors add a reference to the following paper [E. Rej et al. J. Am. Chem. Soc. 2017, 139, 1, 193 - 199 ] (in which the ESR of dangling bonds in nanodiamonds are discussed and shows the g-factors and where ESR lineshapes of P1 and surface radicals overlap).

We will replace the estimation of the $NV^-$ electron spin polarization of the nanodiamonds with those of the microdiamonds (See answer to comment #9). Therefore, we will not discuss the ESR signal of $S = \frac{1}{2}$ spins in the revised version of the SI.

2. Line 9: there are more than three lines here, if the authors include all the features of dangling bonds, one is dealing with multiple overlapping resonances (E. Rej: J. Am. Chem. Soc. 2017, 139, 1, 193 - 199).

See answer to comment #1 of SI.

3. In the supplementary text the authors report T1,C in microdiamonds to be 99 +/- 14 s but then in figure 3(c) the authors show T1,C to be 85+/- 5 s. Please clarify this point.

We thank the reviewer for spotting this typo. The correct T1,C for this measurement is 85+/- 5 s.

**mr-2020-36-SC1**

Since I am interested in DNP, but mostly working in EPR, I have a few questions and comments:

Thank you for your interest in our manuscript.

1. Section 3.2. mentions a cavity. Could you give some details, i.e. the bandwidth? Would frequency sweeps be possible in principle? What is the maximum electron Rabi frequency you can achieve?

Our cavity is TE011 cylindrical cavity resonating at 11.6 GHz. The bandwidth of the cavity is about 5 MHz. In our experiment, the maximum electron Rabi frequency for NV$^-$ center was 11.8 MHz with the input power of 25.5 dBm. For broad frequency sweeps, we would probably need to switch to a broadband resonator and a more powerful MW amplifier (we are currently using a solid state high power amplifier).

2. Section 4.1. The measurement in Figure 3 (b) (optically polarized NV Powder spectrum) took 7h. I have no experience at all with these systems and setups, but I was wondering why it takes so long?

In this measurement in Figure 3, we swept the magnetic field over 0.3 T by taking 900 points, where each point took 0.5 minutes. This lead to a total experimental time of 7 h.

3. Figure 3 (f) I do not think it is absolutely necessary, but since you already use EasySpin to simulate the triplet spectra, it would be straight-forward to include the non-equlibrium populations (either via "Exp.Temperature" in EasySpin version 5 or via "Sys.Pop" in version 6 (developer version)).

Thank you for your suggestion. Figure 3 (f) is used to estimate the active spin-packet fraction $h_p$ that is, the ratio of electron spins that are excited within the magnetic field sweep. Therefore, we believe that it is better to use the thermal spectra to estimate the active spin-packet fraction $h_p$.

mr-2020-36-SC1

Dear Authors,
I very much enjoyed reading your work. In this context and it led to two questions regarding the resonance condition that you have employed.

Thank you for your interest in our manuscript.

First, in your work you appear to have chosen to achieve resonance for perfectly aligned electron spins. Misalignment then results in a loss of resonance and interferes with the polarisation process. In Sec V of Q. Chen, I. Schwarz, F. Jelezko, A. Retzker and M.B. Plenio. Optical hyperpolarization of 13C nuclear spins in nanodiamond ensembles. Phys. Rev. B 92, 184420 (2015) we have also examined an alternative resonance condition chosen for NV orientations that are orthogonal on the external magentic field. Theory suggests that this choice leads to a higher robustness to misorientations as a larger fraction of NVs will be found within a certain range of rotation angles. This was tried out experimentally in J. Scheuer et al. Optically induced dynamic nuclear spin polarization in diamond. New J. Phys. 18, 013040 (2016) but no systematic comparison was done due to the limitation of that particular experiment.

It would be interesting to see whether your eperimental setup may allow for a more careful assessment of the merits of this choice of resonance condition.

Thank you very much for this important input. Increasing the number of crystallites that participate

in DNP is one of the key factors to improve DNP of powdered samples. In our experiments using field-swept ISE only 20 % of all electron spins participate in the DNP process.

Second, in the same work Q. Chen, I. Schwarz, F. Jelezko, A. Retzker and M.B. Plenio. Optical hyperpolarization of 13C nuclear spins in nanodiamond ensembles. Phys. Rev. B 92, 184420 (2015) we have compared a driving scheme in the $|0> <-> |m=+1>$ transition with a scheme that uses a double quantum transition $|m=-1> <-> |0> <-> |m=+1>$ which exhibits a significantly enhanced robustness to misalignments of the NV center. This benefits of this approach have not been tested yet experimentally and it would seem interesting to do so in your current experiment.
Yours sincerely Martin Plenio

Using the double-quantum transition to reduce the orientation dependence of the DNP process is an elegant method for powder samples. As mentioned in the reply to the "short comment #1", higher microwave power might be one challenge for a successful experimental implementation.